# VisCodex: Unified Multimodal Code Generation via Model Merging

**Lingjie Jiang**[1,2]**, Shaohan Huang**[1,*]**Xun Wu**[1]**, Yixia Li**[1,3]**, Guanhua Chen**[3]
**Dongdong Zhang**[1]**, Furu Wei**[1]

[1] Microsoft Research
[2] Peking University
[3] Southern University of Science and Technology
`lingjiejiang@stu.pku.edu.cn; {shaohanh, fuwei}@microsoft.com`

## Abstract

Multimodal large language models (MLLMs) have significantly advanced the integration of visual and textual understanding. However, their ability to generate code from multimodal inputs remains limited. In this work, we introduce VisCodex, a unified framework that seamlessly merges vision and coding language models to empower MLLMs with strong multimodal code generation abilities. Leveraging a task vector-based model merging technique, we integrate a state-of-the-art coding LLM into a strong vision-language backbone, while preserving both visual comprehension and advanced coding skills. To support training and evaluation, we introduce the Multimodal Coding Dataset (McD), a large-scale and diverse collection of 598k samples, including high-quality HTML code, chart image-code pairs, image-augmented StackOverflow QA, and algorithmic problems. Furthermore, we propose InfiBench-V, a novel and challenging benchmark specifically designed to assess models on visually-rich, real-world programming questions that demand a nuanced understanding of both textual and visual contexts. Extensive experiments show that VisCodex achieves state-of-the-art performance among open-source MLLMs and approaches proprietary models like GPT-4o, highlighting the effectiveness of our model merging strategy and new datasets. Our code and data are available at https://github.com/JackLingjie/VisCodex

## 1 Introduction

Multimodal large language models (MLLMs) have achieved remarkable success in recent years, demonstrating an impressive ability to understand and reason about the world by integrating information from both visual and textual domains (Zhu et al., 2023; Liu et al., 2023; Bai et al., 2023). These models have pushed the boundaries of what is possible in tasks like visual question answering (VQA), image captioning, and general multimodal conversation. However, a critical and highly practical domain remains relatively underexplored: the generation of functional code from visual inputs.

This task, which we term multimodal code generation, presents a distinct set of challenges. It demands not only a nuanced interpretation of visual elements—such as UI layouts, data chart structures, or programming-related screenshots—but also the ability to translate these insights into syntactically flawless and functionally correct code. While today's multimodal models excel at visual description, they often lack the deep programming knowledge required for robust code generation. This gap is critical, as many modern development tasks, like translating a UI mockup into HTML or replicating a data chart, demand a seamless fusion of visual understanding and coding proficiency.

To bridge the gap between visual perception and code generation, we introduce VisCodex. Rather than relying on costly pre-training, our approach efficiently creates a unified model by arithmetically merging the parameters of a state-of-the-art vision-language model and a dedicated coding LLM. Specifically, we adopt a model merging technique based on task vectors, which capture the parameter shifts resulting from fine-tuning on specific domains (e.g., vision-language, coding). By linearly

---

[*]Corresponding Author

combining these task vectors in the language model backbone—while keeping the vision encoder and cross-modal projection modules intact—we jointly integrate advanced code understanding and generation capabilities with nuanced visual perception. This enables the resulting model to simultaneously retain strong visual understanding and robust code generation ability, thereby significantly enhancing its performance on multimodal coding tasks. Our experiments show that the merged model significantly outperforms the original vision-language model on multimodal coding tasks.

To address the lack of high-quality, large-scale training data for multimodal code generation, We introduce the **M**ultimodal **c**oding **D**ataset (MCD), a comprehensive, instruction-tuning dataset comprising 598k samples. MCD is meticulously curated from four diverse sources: (1) aesthetically enhanced and structurally sound HTML code generated from webpage screenshots, (2) high-quality chart-to-code pairs from real-world and synthetic sources, (3) image-augmented question-answer pairs from StackOverflow, and (4) foundational algorithmic coding problems to preserve core reasoning abilities.

Furthermore, to rigorously assess the real-world performance of models on multimodal coding QA task, we develop InfiBench-V, a new and challenging benchmark. InfiBench-V consists of visually rich programming-related questions derived from real user scenarios where the images are indispensable for arriving at the correct solution. It provides a more realistic and demanding testbed than existing benchmarks that often focus on either text-only code QA or simpler visual tasks.

Our primary contributions are threefold:

1. We propose VisCodex, a novel approach for creating powerful multimodal code generators by merging vision and coding models, demonstrating a new and efficient path to capability enhancement.
2. We introduce MCD, a large-scale, high-quality dataset for instruction-tuning MLLMs on a wide spectrum of multimodal coding tasks, and InfiBench-V, a challenging benchmark for realistic evaluation. We will release both the dataset and benchmark to facilitate reproducibility and future research.
3. We conduct extensive experiments showing that VisCodex significantly outperforms existing open-source MLLMs and achieves performance competitive with leading proprietary models like GPT-4o, thereby setting a new state of the art for open-source multimodal code generation.

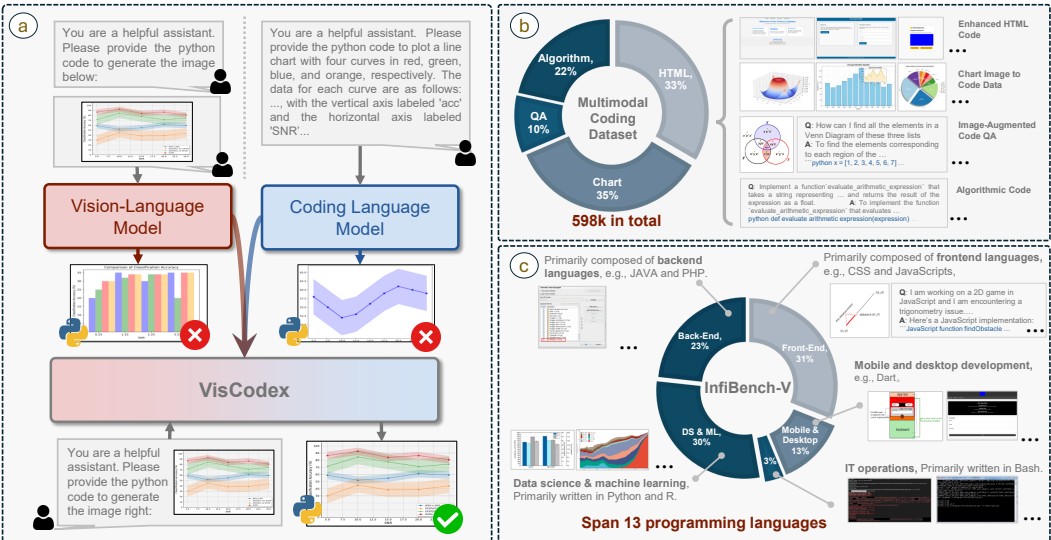

Figure 1: Illustration of the VisCodex pipeline. (a) Model merging strategy for unifying vision-language and coding LLMs; (b) Data distribution and representative cases of MCD; (c) Category breakdown and representative cases of InfiBench-V.

## 2 VISCODEX

### 2.1 MODEL ARCHITECTURE

A typical multimodal large language model (MLLM) is comprised of three primary components: a vision encoder, a language model backbone, and a projection module to connect the two modalities (Li et al., 2024a). The vision encoder's role is to extract visual features from input images. These features are then projected by the projector module into the language embedding space. Subsequently, the language model integrates these visual representations with textual inputs, enabling multimodal understanding and reasoning.

Many existing MLLMs, such as earlier versions of Qwen-VL Bai et al. (2023) and LLaVA Li et al. (2024a), are limited by fixed image input resolutions, which curtails their flexibility in processing images of varying sizes (Bai et al., 2023; Liu et al., 2023). To overcome this limitation, Qwen2.5-VL Wang et al. (2024); Bai et al. (2025) introduces a 2D Rotary Position Embedding (RoPE) (Su et al., 2024) mechanism within its Vision Transformer (ViT) Dosovitskiy et al. (2021). This allows for the flexible processing of images with arbitrary resolutions by dynamically generating visual tokens. This approach preserves the absolute scale and spatial relationships of objects within the image. Given its enhanced flexibility and performance, we adopt the Qwen2.5-VL architecture as our foundation model.

### 2.2 MODEL MERGING

To enhance the coding capabilities of our multimodal large language model without undertaking costly retraining from scratch, we employ model merging (Jin et al., 2022). By arithmetically combining the parameters of specialized models, we can integrate distinct skills and create a unified, more versatile model without requiring access to the original training data (see Figure 1 (a) for an overview of the model merging pipeline).

**Task Vectors.** Central to model merging are *task vectors* (Ilharco et al., 2022), which quantify parameter shifts resulting from fine-tuning a base model on a specific task. Given a pretrained base model $\theta_{\text{base}}$ and its task-specific fine-tuned variant $\theta_{\text{ft}}$, a task vector is formally defined as:

$$\tau_{\text{task}} = \theta_{\text{ft}} - \theta_{\text{base}} \tag{1}$$

Such vectors encapsulate the parameter changes necessary for a model to specialize in a particular domain or capability and serve as modular, transferable units of knowledge across models and tasks.

**Multimodal and Code Capabilities.** Our goal is to enhance the multimodal large language model by incorporating advanced code understanding and generation capabilities. Considering that code-related expertise predominantly resides in the language model backbone, we restrict our merging process to this component. We retain the original visual encoder and cross-modal projection modules unchanged to preserve the intrinsic visual understanding capabilities of the MLLM.

Specifically, we define the task vector for the language model component of the Vision-Language Model (VLM) as:

$$\tau_{\text{vlm}} = \theta_{\text{vlm}} - \theta_{\text{base}} \tag{2}$$

where $\tau_{\text{vlm}}$ captures the parameter shift that enables the language model to effectively handle multimodal inputs by jointly processing visual and textual information.

Analogously, we define the task vector for the coding model, encapsulating its capability for code comprehension and generation:

$$\tau_{\text{code}} = \theta_{\text{code}} - \theta_{\text{base}} \tag{3}$$

**Merging Strategy.** Following the linear merging method of Ilharco et al. (2022) and its application to enhancing multimodal mathematical reasoning in Chen et al. (2025), we adopt a similar strategy to transfer code reasoning abilities into an MLLM. The updated language model parameters, combining both multimodal and code-related knowledge, are computed as follows:

$$\theta_{\text{VisCodex}} = \theta_{\text{base}} + \lambda\tau_{\text{vlm}} + (1 - \lambda)\tau_{\text{code}} \qquad (4)$$

where the hyperparameter $\lambda \in [0, 1]$ controls the trade-off between retaining original multimodal representations and integrating new code expertise. $\theta_{\text{VisCodex}}$ is the initialization of the parameters of our VisCodex.

**Implementation Details.** Our model merging process targets only the language backbone of the VLM, leaving the vision encoder and cross-modal projection modules unaltered. This selective merging approach allows for a clear attribution of performance gains while significantly reducing computational overhead. To construct our primary code task vector ($\tau_{code}$), we select a coding model that shares the same architectural foundation as the VLM's language backbone. Since Qwen2.5-VL's language model is derived from Qwen2.5, we utilize OpenCodeReasoning-Nemotron-1.1-7B (Ahmad et al., 2025). For our 33B scale model, we correspondingly use the OpenCodeReasoning-Nemotron-1.1-32B (Ahmad et al., 2025) variant. Furthermore, in our ablation studies, we create and evaluate code task vectors from two other prominent code-specialized models, Qwen2.5-Coder-7B-Instruct (Hui et al., 2024) and OpenThinker2-7B (Guha et al., 2025), to verify the effectiveness of merging with a code-specialized task vector, as shown in Table 3

## 2.3 Model Training

After model merging, we perform supervised fine-tuning on our Multimodal Coding Dataset (MCD), further aligning the merged model with multimodal coding tasks. To efficiently leverage both the pretrained visual grounding and newly integrated code abilities, we freeze the vision encoder and projection modules, fine-tuning only the language model backbone.

## 3 MCD Dataset

We introduce the Multimodal Coding Dataset (MCD), a new large-scale dataset designed for instruction-tuning multimodal models on coding tasks. MCD is constructed from four primary components, each targeting a distinct aspect of multimodal code understanding and generation:

1. **Enhanced HTML Code**: We generate aesthetically and structurally improved HTML code by redesigning and augmenting existing webpages.
2. **Chart Image-Code Pairs**: We construct high-quality chart-code pairs by sourcing Python matplotlib code from GitHub, which are then refined through a multi-stage filtering and rewriting process.
3. **Image-Augmented Code QA**: We extract real-world, image-augmented question and answer pairs from StackOverflow and subject them to rigorous cleaning and refinement.
4. **Algorithmic Code**: We aggregate and curate data from established algorithmic coding datasets to preserve and enhance the model's core reasoning and problem-solving abilities.

The data distribution and representative cases for the four domains are illustrated in Figure 1 (b), and more comprehensive statistics are available in the Appendix E.1. The following sections describe each component in detail.

### 3.1 Enhanced HTML Code

A review of the existing Web2Code dataset (Yun et al., 2024) revealed several shortcomings, including broken image links, rudimentary CSS, and visually unappealing designs. Our initial approach to address these issues involved using GPT-4o to directly rewrite the existing HTML code. However, this method proved suboptimal, as the constraints of the original code structure frequently led to rendering artifacts and visually incongruous layouts.

To overcome this, we adopted a novel, image-driven generation pipeline. We first curated 560,000 webpage images from Web2Code to serve as stylistic seeds. GPT-4o was then prompted to design entirely new webpages inspired by these seeds. The resulting HTML was rendered using Playwright[1] to capture screenshots. A rigorous filtering pipeline was then applied to discard rendering

---

[1] https://github.com/microsoft/playwright-python

failures, images with anomalous dimensions, and other visual artifacts. This process yielded 200,000 high-quality, newly generated code-image pairs. These pairs were subsequently converted into an instruction-following format using the framework provided by Web2Code.

## 3.2 CHART IMAGE TO CODE DATA

To build a diverse and high-quality chart dataset, we incorporate both synthetic and real-world data sources. For synthetic data, we include the 164,000 synthetic Chart2Code samples released by ChartCoder (Zhao et al., 2025b) as part of our training data. For real-world data, we curated 46,000 chart-code pairs from GitHub.

Inspired by data engineering strategies (Ding et al., 2023; Chiang et al., 2023; Xu et al., 2023), we first collected 129,000 real-world Python matplotlib scripts from GitHub. This raw data, however, suffered from significant quality issues, including non-executable code, inconsistent formatting, and potentially harmful snippets. To mitigate these issues, we employed GPT-4o to systematically rewrite and normalize the code, while simultaneously classifying each script by chart type. Subsequently, a multi-stage, rule-based filtering pipeline was applied to eliminate low-quality samples. This pipeline removed scripts that failed to execute, produced blank or improperly sized images, or generated visually corrupted outputs (e.g., heavily pixelated charts). As a final quality assurance step, we leveraged GPT-4o to score the aesthetic and functional quality of the generated charts, retaining the top 46,000 high-quality image-code pairs.

The final dataset combines the 164,000 synthetic samples with our 46,000 curated real-world examples, resulting in a comprehensive collection of 210,000 chart image-code pairs for instruction tuning.

## 3.3 IMAGE-AUGMENTED CODE QA

StackOverflow represents a rich repository of real-world, code-centric QA data, particularly valuable when augmented with illustrative images. Our collection process involved crawling StackOverflow for QA threads containing images, followed by an initial filtering step to retain only those with an accepted answer containing either Python or HTML code.

A rigorous data cleaning pipeline was implemented to ensure quality, removing entries with excessively short or verbose answers, invalid URLs, broken image links, and blank or oversized images. We also identified that many accepted answers were suboptimal for training, being either too terse for clarity or overly verbose. To address this, we utilized GPT-4o to refine these answers by removing sensitive content, rewriting unclear sections, and enhancing overall conciseness and clarity. This multi-stage pipeline yielded a final dataset of 59,000 high-quality, image-augmented StackOverflow QA pairs suitable for instruction tuning.

## 3.4 ALGORITHMIC CODE

To maintain the model's proficiency in algorithmic reasoning and code generation, we incorporate algorithm-related code data from Kodcode (Xu et al., 2025). Specifically, we select samples from five categories: LeetCode (Hartford, 2023), Codeforces (Jur1cek, 2022), TACO (Li et al., 2023), Code Contests (Li et al., 2022), and Algorithm (The Algorithms, 2023; Keon, 2018). The final collection contains 129,000 algorithm-related instruction-following examples.

## 4 INFIBENCH-V

We introduce InfiBench-V, a new benchmark designed to evaluate the ability of multimodal large language models to answer complex programming questions that integrate both text and images. While existing benchmarks like InfiBench (Li et al., 2024c) focus on text-based code QA, InfiBench-V is specifically constructed to assess multimodal reasoning, where visual context is critical to formulating a correct answer.

## 4.1 DATA CURATION

Our benchmark is built upon a rigorous, multi-stage curation pipeline using data from Stack Overflow. The process began by scraping an initial set of approximately 1 million image-based questions that included a community-verified "accepted answer" to ensure solution quality. We then narrowed this pool to 40,000 recent and high-engagement questions. The most critical refinement step involved using GPT-4o to isolate samples where the image is indispensable, filtering out questions solvable by text alone. This yielded a core set of 10,000 high-relevance, multimodal questions.

We categorized these samples based on programming domain and, guided by the class distribution and sampling principles of InfiBench, domain experts manually selected 322 questions to form the final benchmark. These span 13 programming languages, each mapped to one of five high-level categories: front-end, back-end, data science & machine learning (DS&ML), mobile and desktop development, and IT operations (ITOps). The detailed category breakdown and representative cases are shown in Figure 1 (c)

To ensure quality and prevent model memorization in pre-training, we implemented a prompt paraphrasing process. Domain experts rewrote each question in a concise and directive manner while preserving its semantic content. Each question is also annotated with its evaluation category and associated metrics, including a set of key phrases and a reference answer to support robust scoring.

## 4.2 EVALUATION CRITERIA

To objectively assess the quality of answers across a diverse range of question types, we adopt a three-pronged evaluation strategy inspired by InfiBench. For each benchmark question, domain experts select one or more evaluation methods, and the final score for that question is obtained by averaging the normalized results of the selected methods.

- **Keyword Matching.** We observed that for a majority of questions, answer quality is closely tied to the presence of specific keywords. Our domain experts craft a set of rules for each question, specifying essential terms and phrases. To capture nuanced requirements, these rules can be simple checks, regular expressions, or complex logical statements. When multiple keywords are required, they can be individually weighted to ensure that the most critical components of the answer contribute more significantly to the final score.
- **Unit Testing.** For questions where the answer is primarily a block of code, we verify its correctness using unit tests. To facilitate automated evaluation, domain experts supplement the question with precise requirements, like function names and expected I/O formats. They also provide the necessary setup and teardown scripts, creating a complete and executable environment for programmatic validation.
- **GPT-4o Judge.** For questions that rely heavily on natural language understanding, we leverage GPT-4o to score MLLM responses by comparing them with the accepted reference answer. The evaluation considers both answer correctness and completeness across two dedicated scoring dimensions.

## 5 EXPERIMENTAL SETUP

**Evaluated Benchmarks.** We evaluate our model on four multimodal benchmarks to assess a range of multimodal-related coding skills:

- **Design2Code (Si et al., 2024)**: This benchmark measures the ability to translate visual UI designs into executable code. We report the average performance on both Low-Level (Low-L) features (Block, Text, Position, Color) and High-Level (High-L) semantic fidelity.
- **ChartMimic (Shi et al., 2024)**: This benchmark evaluates the generation of chart specifications from images. We adopt the Direct Mimic task on the test-mini subset and report both Low-Level (Low-L) and GPT-4o-assessed High-Level (High-L) scores.
- **MMCode (Li et al., 2024b)**: This benchmark assesses algorithmic problem-solving in visually rich contexts. Performance is measured by pass@1 accuracy (Chen et al., 2021).
- **InfiBench-V (Ours)**: For our proposed benchmark, we report the average score across all defined evaluation metrics.

**Training Settings.** In our main experiments with the 8B model, which uses the code task vector from OpenCodeReasoning-Nemotron-1.1-7B, we determined the optimal merge coefficient $\lambda$ by evaluating performance on the MMCode benchmark. From a set of candidate values $\{0.7, 0.8, 0.85, 0.9\}$, we selected $\lambda = 0.7$. According to our merging formula 4, this applies a weight of 0.7 to the vision-language task vector ($\tau_{vlm}$) and 0.3 to the code task vector ($\tau_{code}$). Detailed training hyperparameters and training costs are provided in the Appendix C.

# 6 EXPERIMENTAL RESULTS

## 6.1 MAIN RESULTS

Table 1: Performance comparison between proprietary and open-source models across various benchmarks. Low-L stands for Low-Level features (e.g., Block, Text, Position), and High-L stands for High-Level semantic fidelity. Best results are in **bold**.

| Model | Size | Design2Code | | ChartMimic | | MMCode | InfiBench-V | Average |
|---|---|---|---|---|---|---|---|---|
| | | Low-L | High-L | Low-L | High-L | pass@1 | Acc | |
| *Proprietary Models* | | | | | | | | |
| GPT-4o-mini | - | 85.8 | 87.3 | 68.4 | 68.5 | 12.2 | 71.9 | 65.7 |
| GPT-4o | - | 90.2 | 90.4 | 79.0 | 83.5 | 17.0 | 79.9 | 73.3 |
| *Open-Source Small Language Models* | | | | | | | | |
| MiniCPM-V-2_6 | 8B | 78.1 | 84.2 | 21.8 | 45.2 | 3.8 | 45.3 | 46.4 |
| InternVL3-8B | 8B | 85.3 | 87.6 | 43.1 | 47.2 | 6.8 | 66.1 | 56.0 |
| Qwen2.5-VL-7B-Instruct | 8B | 83.4 | 87.6 | 39.5 | 38.3 | 5.3 | 54.0 | 51.4 |
| Llama-3.2-11B-Vision-Instruct | 11B | 72.7 | 84.8 | 27.7 | 26.5 | 2.3 | 52.7 | 44.4 |
| InternVL3-14B | 15B | 82.9 | 88.3 | 53.9 | 55.0 | **11.4** | 70.5 | 60.3 |
| **VisCodex-8B** | **8B** | **90.1** | **90.9** | **74.8** | **74.1** | 11.0 | **72.1** | **68.8** |
| *Open-Source Large Language Models* | | | | | | | | |
| Qwen2.5-VL-32B-Instruct | 33B | 88.0 | 89.4 | 72.5 | 68.7 | 13.7 | 73.0 | 67.6 |
| llava-onevision-qwen2-72b | 73B | 75.2 | 85.7 | 55.8 | 52.1 | 5.7 | 64.7 | 56.5 |
| Qwen2.5-VL-72B-Instruct | 73B | 86.9 | 88.7 | 66.7 | 68.7 | 15.2 | 75.2 | 66.9 |
| InternVL3-78B | 78B | 85.3 | 89.1 | 64.9 | 64.2 | 14.4 | 77.3 | 65.9 |
| **VisCodex-33B** | **33B** | **90.5** | **91.1** | **79.3** | **78.5** | **15.6** | **78.6** | **72.3** |

As shown in Table 1, our models achieve state-of-the-art performance across all evaluated multimodal coding benchmarks. Our smaller model, VisCodex-8B, not only outperforms all open-source models in its size class (7-15B) but also surpasses the proprietary GPT-4o-mini, with an average score of 68.8. Our larger model, VisCodex-33B, further establishes its superiority by achieving an average score of 72.3, which is on par with the state-of-the-art proprietary model, GPT-4o (73.3). These results demonstrate that our VisCodex family sets a new standard for open-source multimodal code generation.

Our models show exceptional strength in UI and chart understanding. On the Design2Code benchmark, both VisCodex-8B (90.1/90.9) and VisCodex-33B (90.5/91.1) achieve scores comparable to or exceeding GPT-4o. On ChartMimic, our models also secure the top positions among open-source models, demonstrating robust visual data translation capabilities.

## 6.2 ANALYSIS

**Efficacy of the Model Merging.** As demonstrated in Table 2, model merging yields consistent performance gains across all benchmarks and scales. At the 8B scale, merging improves Design2Code (90.1 vs. 89.6), ChartMimic (74.8 vs. 73.4), and MMCode (11.0 vs. 6.8). The 33B model shows similar enhancements. The most significant improvements on ChartMimic and MMCode confirm that this strategy effectively augments code-generation capabilities while preserving visual understanding.

Table 2: Ablation on model merging for VisCodex. "w/o model merge" denotes the variant without applying our model merging strategy.

| Method | Design2Code | | ChartMimic | | MMCode |
|---|---|---|---|---|---|
| | Low-L | High-L | Low-L | High-L | pass@1 |
| **VisCodex-8B** | **90.1** | **90.9** | **74.8** | **74.1** | **11.0** |
| w/o model merge | 89.6 | 90.7 | 73.4 | 70.6 | 6.8 |
| **VisCodex-33B** | **90.5** | **91.1** | **79.3** | **78.5** | **15.6** |
| w/o model merge | 89.7 | 90.7 | 78.4 | 77.4 | 14.4 |

Table 3: Ablation on Backbone LLM choice. Performance when merging the same multimodal backbone with either a general-purpose or code-specialized LLM.

| Backbone LLM | Design2Code | | ChartMimic | | MMCode |
|---|---|---|---|---|---|
| | Low-L | High-L | Low-L | High-L | pass@1 |
| Baseline (Qwen2.5-VL) | 83.4 | 87.6 | 39.5 | 38.3 | 5.3 |
| *General LLM* | | | | | |
| Qwen2.5-7B-Instruct | 89.5 | 90.7 | 73.2 | 72.5 | 6.8 |
| *Code LLM* | | | | | |
| OpenThinker2-7B | 90.2 | 91.0 | 74.3 | 73.8 | 8.0 |
| Qwen2.5-Coder-7B | 90.0 | 90.7 | 75.1 | 74.5 | 8.4 |
| Nemotron-1.1-7B | 90.1 | 90.9 | 74.8 | 74.1 | 11.0 |

Table 4: Performance comparison of model merging vs. backbone replacement. The "Replace (1-stage)" strategy directly replaces the LLM backbone in a single stage. The "Replace (2-stage)" strategy first trains a projector, then fine-tunes the full MLLM (ViT, projector, and LLM).

| Strategy | Design2Code | | ChartMimic | | MMCode |
|---|---|---|---|---|---|
| | Low-L | High-L | Low-L | High-L | pass@1 |
| Baseline | 83.4 | 87.6 | 39.5 | 38.3 | 5.3 |
| Replace (1-stage) | 88.7 | 90.7 | 70.4 | 69.2 | 11.0 |
| Replace (2-stage) | 88.2 | 90.6 | 73.4 | 70.9 | 11.0 |
| **Model Merge (Ours)** | **90.1** | **90.9** | **74.8** | **74.1** | **11.0** |

**Effect of Different Code LLMs in Merge.** As shown in the Table 3, we study how the choice of the merged LLM affects performance. All code-specialized LLMs present consistent gains across all benchmarks compared to general-purpose LLM. Compared to the general LLM, OpenThinker2-7B and Qwen2.5-Coder-7B improve both Design2Code and ChartMimic, while Nemotron-1.1-7B further boosts MMCode pass@1 from 6.8 to 11.0. These results indicate that merging with code-specialized LLMs is crucial for robust multimodal code generation, enhancing executable correctness while maintaining strong visual grounding and UI-to-code translation.

**Effectiveness of the Model Merge Strategy.** To evaluate the effectiveness of our proposed model merge strategy compared to direct backbone replacement, we conducted comparative experiments using two distinct approaches: (i) directly replacing the LLM backbone of Qwen2.5-VL-7B-Instruct with OpenCodeReasoning-Nemotron-1.1-7B (Ahmad et al., 2025), and (ii) employing the two-stage training procedure from LLaVA-OneVision (Li et al., 2024a), which initially trains the projector on BLIP-558K, followed by joint fine-tuning of the ViT, projector, and LLM on MCD.

Our results indicate that the model merge strategy achieves overall superior performance across the evaluated tasks, as shown in Table 4. It demonstrates particularly strong gains on visually-intensive benchmarks such as Design2Code and ChartMimic, where successful code generation heavily relies on accurate visual-semantic alignment. This is because directly replacing the LLM backbone often disrupts previously learned visual grounding. In contrast, the model merge approach preserves these visual alignment abilities while simultaneously incorporating enhanced code generation capabilities. This confirms the effectiveness of model merging in maintaining multimodal comprehension and boosting performance in multimodal coding tasks.

**Additional Analyses.** Further results are provided in the Appendix, including comparisons with existing Web2Code datasets (Appendix B.1), the generalizability of MCD (Appendix B.2), and the generality of our code model merging strategy (Appendix B.3).

### 6.3 CASE STUDY

We further conducted case studies to qualitatively compare the performance of VisCodex-8B against GPT-4o, InternVL3-78B, and Qwen2.5-VL-7B on the ChartMimic and Design2Code benchmarks. As shown in Figure 2, VisCodex-8B consistently generates outputs that more closely match the ground truth in both chart reconstruction and HTML generation tasks, surpassing the fidelity of results produced by GPT-4o-mini and other open-source baselines. These observations underscore VisCodex-8B's superior multimodal code generation capabilities. For additional case studies on MMCode, InfiBench-V, and further examples, please refer to Appendix H.

## 7 RELATED WORK

### 7.1 MULTIMODAL CODE GENERATION

The ability of MLLMs to generate code has attracted increasing attention in recent years. Design2Code (Si et al., 2024) evaluates the HTML generation capabilities of MLLMs. Extending earlier datasets like WebSight (Laurençon et al., 2024) and Pix2Code (Beltramelli, 2018), Web2Code (Yun et al., 2024), Webcode2M Gui et al. (2025) provides a webpage-to-code dataset to improve HTML

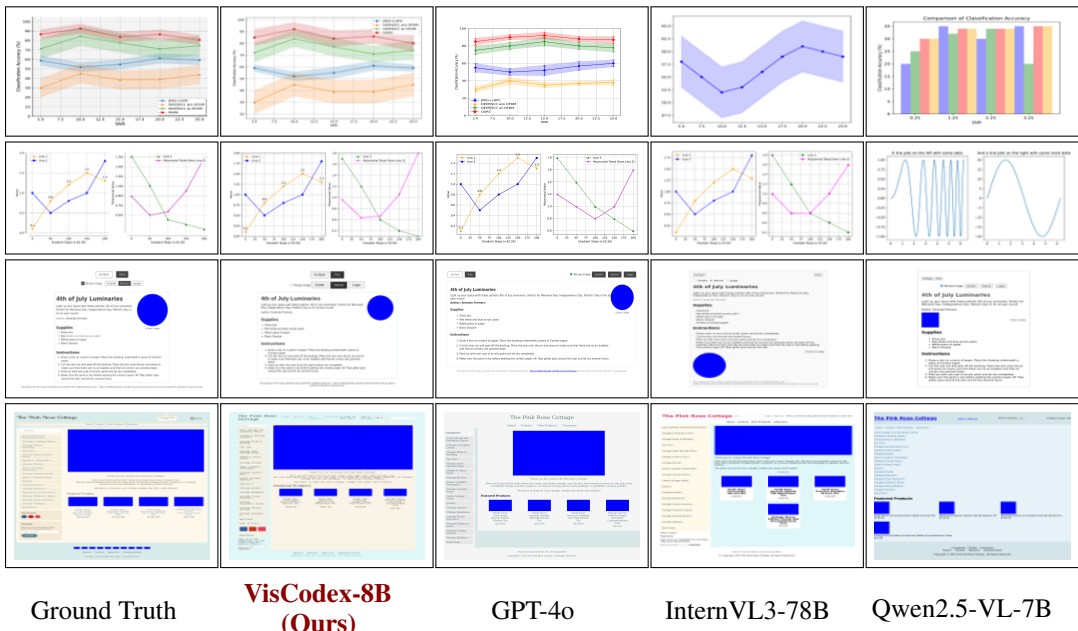

|                  |                          |          |                 |                 |
|------------------|--------------------------|----------|-----------------|-----------------|
| Ground Truth     | **VisCodex-8B (Ours)**   | GPT-4o   | InternVL3-78B   | Qwen2.5-VL-7B   |

Figure 2: Case study comparing VisCodex-8B and baseline models on Design2Code and ChartMimic tasks, demonstrating the superior multimodal code generation capabilities of VisCodex-8B.

generation. Benchmarks like MMCode (Li et al., 2024b) and Human-V (Zhang et al., 2024b) focus on assessing MLLMs in algorithmic coding tasks that incorporate visual inputs. Similarly, Chart-Mimic (Shi et al., 2024) and Plot2Code (Wu et al., 2024) evaluate MLLMs' capabilities to translate raw data into scientific charts. ChartCoder (Zhao et al., 2025a) addresses chart generation explicitly through a large dataset of 160k examples. Additionally, CodeV (Zhang et al., 2024c) integrates visual data to improve large language models' problem-solving abilities. Despite recent progress, to the best of our knowledge, existing work falls short of providing a complete and unified solution to multimodal code generation.

### 7.2 MODEL MERGING FOR MLLMS

Model merging has become a widely used approach for integrating the capabilities of multiple models within the parameter space. A basic method involves simple weighted averaging (Wortsman et al., 2022), while more advanced strategies have been developed in recent years (Ilharco et al., 2022; Matena & Raffel, 2022; Jin et al., 2022; Yadav et al., 2023; Bandarkar et al., 2024). Recently, several studies have applied model merging to enhance the capabilities of multimodal large language models. For example, REMEDY (Zhu et al., 2025a) improves multitask performance and zero-shot generalization in VQA tasks. (Akiba et al., 2025) enhance Japanese language understanding and generation, while Chen et al. (2025) improve mathematical reasoning abilities. Li et al. (2025b) enable textual preference transfer by integrating a text-based reward model into an MLLM, without additional training. Our study demonstrates that model merging can effectively endow MLLMs with strong abilities in multimodal code understanding and generation.

### 8 CONCLUSION

In conclusion, we have presented VisCodex, a unified multimodal framework that effectively integrates advanced visual comprehension with sophisticated code-generation capabilities through a novel task vector-based model merging strategy. By leveraging this efficient approach, VisCodex significantly enhances multimodal large language models without incurring the costs associated with full-scale retraining. We also introduced the Multimodal Coding Dataset (MCD), a comprehensive resource comprising 598k diverse, high-quality instruction-tuning examples, along with InfiBench-V, a rigorous benchmark designed specifically for realistic multimodal coding assessments. Extensive

experiments confirm that VisCodex establishes a new state-of-the-art performance among open-source multimodal code generators, demonstrating capabilities competitive with leading proprietary models such as GPT-4o.

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

# A    LLM USAGE STATEMENT

A large language model (ChatGPT) was used to **aid and polish the writing of the paper**, including minor grammar correction and language refinement.

# B    ADDITIONAL ANALYSIS EXPERIMENTS

## B.1    COMPARISON WITH EXISTING WEB2CODE DATASETS

As demonstrated in Table 5, our dataset outperforms prior Web2Code corpora across all metrics. Compared to Webcode2M Gui et al. (2025) and Web2Code Yun et al. (2024), MCD achieves the highest scores on both low-level (Block-Match, Text, Position, and Color) and high-level evaluation. The gains are especially notable on layout-sensitive metrics and visual fidelity, indicating that MCD provides more accurate structural alignment and visual grounding for UI-to-code generation, while also improving semantic consistency.

Table 5: Performance comparison of MCD with WebCode2M and Web2Code on the Design2Code benchmark.

| Data | Block-Match | Text | Position | Color | CLIP |
|------|-------------|------|----------|-------|------|
| Baseline | 85.4 | 95.8 | 77.3 | 75.3 | 87.6 |
| WebCode2M | 82.1 | 96.0 | 72.4 | 72.6 | 86.3 |
| Web2Code | 84.4 | 93.4 | 76.2 | 79.6 | 88.8 |
| MCD | **89.6** | **97.2** | **84.7** | **86.8** | **90.7** |

## B.2    GENERALIZABILITY OF MCD

To assess the generalizability of our dataset MCD, we conducted supervised fine-tuning experiments on two strong open-source MLLMs: **InternVL3-8B** (Zhu et al., 2025b) and **llava-llama3.1-8b**[2] (Zhang et al., 2024a). We evaluated the models on multiple established benchmarks, as shown in Table 6. The results demonstrate that fine-tuning with MCD consistently and significantly enhances the multimodal coding abilities of both base models. In particular, we observe substantial improvements in both the **Design2Code** and **ChartMimic** tasks, as well as noticeable gains on the **MMCode** benchmark. These findings validate the robustness and strong transferability of MCD across different model architectures and suggest its value as a general-purpose resource for advancing multimodal code generation.

Table 6: Generalization performance of MCD: Results of supervised fine-tuning on InternVL3-8B and llava-llama3.1-8b across multiple multimodal coding benchmarks.

| Model | Design2Code | | ChartMimic | | MMCode |
|-------|------|------|------|------|---------|
| | Low | High | Low | High | pass@1 |
| InternVL3-8B | 85.3 | 87.6 | 43.1 | 46.6 | 6.8 |
| **InternVL3-8B-SFT** | **88.2** | **89.9** | **72.6** | **70.4** | **7.6** |
| llava-llama3.1-8b | 7.3 | 78.4 | 6.2 | 4.8 | 2.3 |
| **llava-llama3.1-8b-SFT** | **82.8** | **90.5** | **70.7** | **68.2** | **4.2** |

## B.3    GENERALITY OF CODE MODEL MERGING

To verify that our model merging strategy is not limited to LLMs with Qwen2.5 backbones, we further conduct experiments using llava-llama3.1-8b (Zhang et al., 2024a), an MLLM based on the Llama-3.1 (Dubey et al., 2024) architecture. Specifically, we merge llava-llama3.1-8b with the code task vector ($\tau_{code}$) obtained from DeepSeek-R1-Distill-Llama-8B (DeepSeek-AI, 2025), following

---

[2]https://huggingface.co/modelscope/llava-llama3.1-8b

the same procedure as in our previous Qwen2.5VL experiments. As shown in Table 7, the merged model consistently outperforms the non-merged baseline across all multimodal coding tasks. The performance improvement is particularly significant on the MMCode benchmark, where pass@1 increases from 4.2 to 6.5. These results, which are consistent with our findings on Qwen2.5VL, demonstrate the general effectiveness and broad applicability of the code model merging strategy.

Table 7: Ablation on model merging for **llava-llama3.1-8b** and **DeepSeek-R1-Distill-Llama-8B**. Results compare models with and without the merging strategy.

| Method | Design2Code | | ChartMimic | | MMCode |
|---|---|---|---|---|---|
| | Low-L | High-L | Low-L | High-L | pass@1 |
| w/o model merge | 82.8 | 90.5 | 70.7 | 68.2 | 4.2 |
| w model merge | **83.3** | **90.7** | **70.9** | **68.8** | **6.5** |

## B.4 POST-MERGE UNFREEZING ABLATION

To further investigate whether weight interpolation introduces any cross-modal distribution shift after model merging, we conduct additional ablation studies by unfreezing different subsets of visual modules during post-merge supervised fine-tuning. Specifically, we compare three tuning strategies: (1) tuning only the LLM backbone (our default approach), (2) additionally unfreezing the cross-modal projector, and (3) unfreezing both the projector and the ViT encoder.

As shown in Table 8, unfreezing visual components does not lead to consistent performance improvements across benchmarks. All variants yield comparable results on Design2Code, ChartMimic, and MMCode. This indicates that the merged model maintains robust cross-modal alignment without requiring further adaptation of visual modules. Consequently, our parameter-efficient strategy of tuning only the LLM backbone remains both effective and computationally preferable.

Table 8: **Post-merge Unfreezing Ablation.** Comparison of different tuning scopes. We report the performance when training specific components while keeping others frozen. **LLM (VisCodex-8B)** represents our default strategy.

| Tunable Modules | Design2Code | | ChartMimic | | MMCode |
|---|---|---|---|---|---|
| | Low-level | High-level | Low-level | High-level | pass@1 |
| Baseline (Qwen2.5-VL-7B-Instruct) | 83.4 | 87.6 | 39.5 | 38.3 | 5.3 |
| **LLM (VisCodex-8B)** | **90.1** | **90.9** | **74.8** | 74.1 | **11.0** |
| LLM + Projector | 89.9 | 90.7 | 73.4 | 74.1 | 9.5 |
| LLM + Projector + ViT | **90.1** | **90.9** | 74.6 | **74.7** | 10.6 |

## B.5 COMPARISON WITH LoRA FINE-TUNING STRATEGIES

To further validate the superiority of our Model Merging strategy over standard parameter-efficient adaptation methods, and to investigate the impact of tuning different modules, we conducted a comprehensive comparison with LoRA (Hu et al., 2022) fine-tuning. We established two distinct baselines:

- **LoRA (LLM):** Applying LoRA tuning exclusively to the language model backbone, serving as a parameter-efficient counterpart to our language-only tuning strategy.
- **LoRA (ViT + Proj + LLM):** Jointly tuning the vision encoder, projector, and language model using LoRA. This setting was designed to test whether broader parameter adaptation (i.e., including the vision encoder) could bridge the performance gap.

As shown in Table 9, VisCodex significantly outperforms all LoRA baselines, even when the vision encoder is tuned (*ViT + Proj + LLM*), with substantial margins of **9.0%** on ChartMimic (Low-L) and **4.2%** on MMCode. While broader vision tuning offers slight gains over *LLM-only* LoRA, it still fails to match VisCodex. Crucially, since both *SFT (Full)* and *VisCodex* employ the same full fine-tuning strategy, our superior performance confirms that **Model Merging initialization** provides a robust "knowledge injection" of code capabilities that standard SFT initialization cannot replicate.

Table 9: Comparison of VisCodex with Full Fine-tuning and LoRA tuning strategies on the MCD dataset. VisCodex consistently outperforms baselines, demonstrating the effectiveness of model merging initialization combined with full parameter tuning.

| Method | Strategy | Tunable Modules | Design2Code Low-L | Design2Code High-L | ChartMimic Low-L | ChartMimic High-L | MMCode pass@1 |
|---|---|---|---|---|---|---|---|
| *Base Model* | | | | | | | |
| Qwen2.5-VL-7B-Instruct | - | - | 83.4 | 87.6 | 39.5 | 38.3 | 5.3 |
| *Standard Initialization (w/o Model Merge)* | | | | | | | |
| SFT (LoRA) | LoRA | LLM | 87.7 | 89.2 | 64.0 | 65.1 | 5.7 |
| SFT (LoRA) | LoRA | ViT + Proj + LLM | 88.2 | 89.6 | 65.8 | 67.0 | 6.8 |
| SFT (Full) | Full FT | LLM | 89.6 | 90.7 | 73.4 | 70.6 | 6.8 |
| *Ours (w/ Model Merge)* | | | | | | | |
| **VisCodex-8B** | **Full FT** | **LLM** | **90.1** | **90.9** | **74.8** | **74.1** | **11.0** |

## B.6 DATA EFFICIENCY AND ROBUSTNESS IN LOW-RESOURCE SETTINGS

To investigate the potential of model merging in data-scarce scenarios and its efficiency in transferring prior knowledge, we conducted two sets of additional experiments: (1) training with strictly limited data volumes (Data Scaling), and (2) training without specific algorithmic reasoning data (Domain-Specific Scarcity).

**Performance under Data-Scarce Conditions.** To evaluate the model's performance when overall training data is limited, we trained VisCodex-8B on randomly sampled subsets of the MCD dataset: 1% (6k samples), 5% (30k samples), and 10% (60k samples). We compared our model merging strategy against the standard fine-tuning baseline (w/o model merge).

Table 10 presents the results. Our model merging strategy consistently outperforms the baseline across all data scales. Notably, in the extremely low-resource setting (1% data), the standard approach suffers from catastrophic forgetting in reasoning tasks, with MMCode performance dropping significantly from 5.3 (Zero-shot Baseline) to 3.4. In contrast, VisCodex effectively retains the baseline reasoning capability (5.3) due to the strong prior injected via model merging. Furthermore, even with only 5% of the data (30k), VisCodex achieves robust performance improvements, highlighting the significant data efficiency gained from the merged code priors.

Table 10: Performance comparison on limited training data (Data Scaling). We report the results of training on 1%, 5%, and 10% subsets of the MCD dataset.

| Data Scale | Method | Design2Code Low-L | Design2Code High-L | ChartMimic Low-L | ChartMimic High-L | MMCode pass@1 |
|---|---|---|---|---|---|---|
| 0% (Baseline) | Qwen2.5-VL-7B-Instruct | 83.4 | 87.6 | 39.5 | 38.3 | 5.3 |
| 1% (6k) | w/o model merge | 87.9 | 89.5 | 60.1 | 62.1 | 3.4 |
| | **w/ model merge (Ours)** | **89.2** | **89.5** | **61.6** | **63.0** | **5.3** |
| 5% (30k) | w/o model merge | 88.9 | 89.8 | 68.2 | 68.6 | 3.8 |
| | **w/ model merge (Ours)** | **89.1** | **89.8** | **70.2** | **70.7** | **5.3** |
| 10% (60k) | w/o model merge | 89.4 | 90.1 | 69.5 | 69.1 | 3.8 |
| | **w/ model merge (Ours)** | **89.8** | **90.4** | **70.6** | **71.0** | **7.6** |

**Efficiency in Domain-Specific Data Scarcity.** To simulate a scenario where task-specific data is unavailable, we conducted an ablation study by removing the entire "Algorithm" category (129k samples) from the MCD dataset. Crucially, the "Algorithm" category in MCD consists of text-only code problems. Since MCD does not contain multimodal algorithmic data to begin with, removing this category implies that the model is fine-tuned without any exposure to algorithmic reasoning data (neither text-only nor multimodal).

As shown in Table 11, when explicit algorithmic training data is absent, the standard SFT approach (w/o model merge) fails to generalize to the multimodal reasoning task (MMCode), resulting in a low

score of 3.4. However, VisCodex achieves a score of 6.8 even without seeing any algorithmic training samples. These results validate that our model merging strategy significantly enhances efficiency by leveraging injected code reasoning capabilities, making the model robust even when specific domain data is entirely absent.

Table 11: Ablation study on removing "Algorithm" training data (Efficiency in Domain-Specific Scarcity). The models are trained on the MCD dataset excluding the Algorithm category.

| Method | Design2Code | | ChartMimic | | MMCode |
|---|---|---|---|---|---|
| | Low-L | High-L | Low-L | High-L | pass@1 |
| w/o model merge | 89.3 | 90.3 | 73.2 | 70.5 | 3.4 |
| **w/ model merge (Ours)** | **90.0** | **90.4** | **74.2** | **74.6** | **6.8** |

### B.7 STATISTICAL SIGNIFICANCE OF PERFORMANCE GAINS AT SCALE

To address concerns regarding whether the performance improvements observed in larger models (specifically the 33B variant) are statistically significant or merely attributable to random variance, we conducted a rigorous robustness analysis. We performed 5 independent inference runs for both the merged model and the standard fine-tuned baseline across all benchmarks. To account for generation stochasticity, we utilized a sampling temperature of $T = 0.6$.

The results, summarized in Table 12, demonstrate that the proposed model merging method consistently outperforms the baseline across all evaluated metrics. The performance gains range from $+0.5$ to $+1.4$ points, with low standard deviations ($\sigma \leq 0.29$), indicating high stability. Furthermore, we conducted paired $t$-tests to quantify significance. The resulting $p$-values range from $1.4 \times 10^{-4}$ to $2.6 \times 10^{-7}$, which are orders of magnitude below the conventional $\alpha = 0.05$ threshold. These findings confirm that the advantages of model merging remain robust and statistically significant even at the 33B scale.

Table 12: **Statistical significance analysis of the 33B model across 5 independent runs** ($T = 0.6$). We report the Mean $\pm$ Std. The "w/ model merge" method consistently outperforms the baseline with statistically significant gains across all benchmarks ($p < 0.05$).

| Method | Design2Code Low | Design2Code High | ChartMimic Low | ChartMimic High | MMCode pass@1 |
|---|---|---|---|---|---|
| w/o model merge | $89.68 \pm 0.08$ | $90.62 \pm 0.08$ | $78.36 \pm 0.11$ | $77.32 \pm 0.13$ | $14.38 \pm 0.04$ |
| w/ model merge | $\mathbf{90.56 \pm 0.05}$ | $\mathbf{91.14 \pm 0.05}$ | $\mathbf{79.74 \pm 0.29}$ | $\mathbf{78.66 \pm 0.11}$ | $\mathbf{15.68 \pm 0.18}$ |
| *p-value* | $2.6 \times 10^{-7}$ | $8.8 \times 10^{-6}$ | $1.4 \times 10^{-4}$ | $1.5 \times 10^{-7}$ | $4.2 \times 10^{-5}$ |

### B.8 NECESSITY OF VISUAL GROUNDING: COMPARISON WITH STANDALONE CODE LLMS

To investigate whether the performance gains of VisCodex stem primarily from the inherent strength of the code branch rather than true multimodal fusion, we evaluated several state-of-the-art 7B-scale Code LLMs as **standalone agents** across our multimodal benchmarks. Specifically, we tested Qwen2.5-Coder-7B-Instruct, OpenCodeReasoning-Nemotron-7B, and OpenThinker2-7B using the same input prompts (text description + image placeholders) as the multimodal models.

As shown in Table 13, despite their exceptional text-only programming capabilities, these standalone Code LLMs perform near-zero on tasks requiring visual grounding. The near-zero scores are **not** due to a lack of coding ability, but rather because these models **lack a visual encoder** and cannot process the essential image inputs.

- For instance, on DESIGN2CODE and CHARTMIMIC, which heavily rely on interpreting visual UI elements and chart data, the code models fail to generate meaningful outputs, achieving negligible scores (e.g., 0.0 on DESIGN2CODE Low-Level metrics).

- Similarly, on INFIBENCH-V, the accuracy hovers around 1.2%–1.5%, confirming the models cannot solve the problems **without visual context**.

In contrast, the base VLM (Qwen2.5-VL-7B-Instruct) demonstrates reasonable visual understanding (83.4 on DESIGN2CODE) but lacks advanced code reasoning capabilities (5.3 on MMCODE). VisCodex-8B significantly outperforms both the base VLM and the standalone Code LLMs across all metrics. This empirical evidence confirms that neither visual ability alone nor code ability alone is sufficient for these tasks; the performance improvements are driven by the effective fusion of visual perception and code reasoning within our framework.

Table 13: Performance comparison of VisCodex against standalone Code LLMs. "Low" refers to Low-level visual metrics (Block/Text/Position/Color match), and "pass@1" refers to reasoning accuracy. The results demonstrate that strong code models alone cannot solve multimodal tasks without visual grounding.

| Model | Type | Design2Code (Low-Level) | ChartMimic (Low-Level) | MMCode (pass@1) | InfiBench-V (Acc) |
|---|---|---|---|---|---|
| Qwen2.5-Coder-7B-Instruct | Code LLM | 0.0 | 6.1 | 6.1 | 1.5 |
| OpenCodeReasoning-Nemotron-7B | Code LLM | 0.2 | 2.8 | 1.5 | 1.2 |
| OpenThinker2-7B | Code LLM | 0.0 | 2.4 | 1.9 | 1.2 |
| Qwen2.5-VL-7B-Instruct | Base VLM | 83.4 | 39.5 | 5.3 | 54.0 |
| **VisCodex-8B (Ours)** | **Merged VLM** | **90.1** | **74.8** | **11.0** | **72.1** |

## B.9 COMPARISON WITH SPECIALIZED CHART MODEL

To further validate the effectiveness of VisCodex against specialized multimodal code models, we conducted a direct comparative analysis with ChartCoder (Zhao et al., 2025a), a state-of-the-art model explicitly optimized for chart-to-code generation. As shown in Table 14, VisCodex-8B consistently outperforms the specialized ChartCoder across all evaluated benchmarks.

Even on the domain-specific ChartMimic benchmark, where ChartCoder is specifically tuned, VisCodex achieves superior performance (74.8 vs. 72.5 on Low-Level metrics and 74.1 vs. 74.0 on High-Level metrics). Furthermore, on generalized tasks such as Design2Code and InfiBench-V, VisCodex demonstrates a significant advantage (e.g., 90.1 vs. 36.7 on Design2Code Low-Level). These results highlight the core advantage of VisCodex: it functions as a unified multimodal code generator that matches or exceeds the performance of domain-specific specialists while maintaining robust versatility across diverse coding tasks.

Table 14: Performance comparison between ChartCoder and VisCodex-8B across multimodal coding benchmarks. VisCodex outperforms the specialist model on its specific domain (ChartMimic) while maintaining significantly higher performance on general tasks.

| Model | Design2Code | | ChartMimic | | MMCode | InfiBench-V |
|---|---|---|---|---|---|---|
| | Low-level | High-level | Low-level | High-level | pass@1 | Acc. |
| ChartCoder *(Specialist)* | 36.7 | 82.7 | 72.5 | 74.0 | 2.7 | 32.3 |
| **VisCodex-8B *(Ours)*** | **90.1** | **90.9** | **74.8** | **74.1** | **11.0** | **72.1** |

## B.10 SENSITIVITY ANALYSIS OF MERGE COEFFICIENT $\lambda$

To address concerns regarding the heuristic selection of the balancing factor $\lambda$ and to evaluate the robustness of our model merging strategy, we conducted a comprehensive sensitivity analysis. We evaluated the performance of VisCodex-8B across a range of mixing coefficients $\lambda \in \{0.0, 0.7, 0.8, 0.85, 0.9\}$. Recall that according to Equation 4, $\lambda$ controls the weight of the vision-language task vector ($\tau_{vlm}$), while $(1 - \lambda)$ controls the coding task vector ($\tau_{code}$).

As presented in Table 15, the merged model exhibits a high degree of robustness rather than relying on a narrow heuristic optimum:

- **Broad Performance Plateau:** For $\lambda$ values between 0.7 and 0.9, visual understanding capabilities (measured by Design2Code and ChartMimic) remain consistently high, with minimal

variance. This suggests that the semantic directions of the two task vectors do not destructively interfere.

- **Reasoning vs. Vision Trade-off:** As $\lambda$ decreases (increasing the influence of $\tau_{code}$), we observe a steady improvement in algorithmic reasoning, with MMCode pass@1 increasing from 7.2 to 11.0.
- **Necessity of Multimodal Alignment:** Critically, setting $\lambda = 0.0$ (effectively using only the code adaptation shift) results in a significant performance drop on visually intensive tasks (e.g., ChartMimic average score drops from 74.5 to 69.8). This confirms that the VLM task vector ($\tau_{vlm}$) provides essential multimodal alignment that cannot be supplied by the code model alone.

Table 15: Sensitivity analysis of the merge coefficient $\lambda$ on VisCodex-8B performance. The "Avg" columns represent the average of Low-Level and High-Level scores for the respective benchmarks.

| $\lambda$ | Design2Code (Avg) | ChartMimic (Avg) | MMCode (pass@1) |
|------|------|------|------|
| 0.9 | **90.6** | 73.5 | 7.2 |
| 0.85 | 90.5 | 74.1 | 7.6 |
| 0.8 | 90.5 | 74.2 | 8.0 |
| 0.7 | 90.5 | **74.5** | **11.0** |
| 0.0 | 89.7 | 69.8 | 11.0 |

### B.11 Robustness Analysis under Visual Degradation

To assess the robustness of VisCodex under degraded visual conditions, we constructed a **Composite Noisy Dataset** derived from the Design2Code benchmark. For each sampled instance, we randomly applied either Gaussian Blur or Low-Resolution Downsampling to simulate realistic low-quality images found in real-world scenarios. We evaluated both our model and the baseline under this mixed-noise setting.

As shown in Table 16, VisCodex-8B exhibits stronger resilience to noise compared to the baseline. While the baseline Qwen2.5-VL-7B-Instruct loses 0.8 points on High-Level metrics, VisCodex drops only 0.2 points. Furthermore, on Low-Level metrics, our model maintains a high score of 89.6 even under noisy conditions, significantly outperforming the baseline's original performance (83.4). These results indicate that merging code priors does not weaken visual robustness; rather, it appears to stabilize structural inference even when visual details are degraded.

Table 16: Robustness analysis on the Composite Noisy Dataset (Design2Code). The "Noisy" condition includes random application of Gaussian Blur or Downsampling. $\Delta$ indicates the performance drop under noisy conditions.

| Model | Condition | Low-Level | High-Level |
|------|------|------|------|
| Qwen2.5-VL-7B-Instruct | Original | 83.4 | 87.6 |
| Qwen2.5-VL-7B-Instruct | Noisy | 82.7$_{-0.7}$ | 86.8$_{-0.8}$ |
| VisCodex-8B | Original | 90.1 | 90.9 |
| VisCodex-8B | Noisy | 89.6$_{-0.5}$ | 90.7$_{-0.2}$ |

### B.12 Impact on General Visual Question Answering

A potential concern with model merging is catastrophic forgetting, where enhancing code capabilities might degrade general visual understanding. To evaluate this, we assessed the model on three general vision-language benchmarks: TextVQA (Singh et al., 2019), ChartQA (Masry et al., 2022), and GQA (Hudson & Manning, 2019).

As presented in Table 17, the performance differences between VisCodex and the baseline are minimal ($< 1.3$ on TextVQA and $< 0.8$ on GQA) and fall within typical variance observed in model merging studies. This indicates no meaningful catastrophic forgetting. Crucially, these minor fluctuations are acceptable trade-offs given the substantial absolute gains achieved across multimodal code-generation tasks (e.g., +6.7 points on Design2Code, +35.3 points on ChartMimic, and +5.7 points on MMCode).

It is also worth noting that while general TextVQA drops slightly, domain-specific UI text recognition (measured by Design2Code Low-Level metrics) actually improves significantly from 83.4 to 90.1.

Table 17: Assessment of Catastrophic Forgetting on general VQA benchmarks. The slight decreases in performance are negligible compared to the significant gains in coding tasks.

| Model | TextVQA | ChartQA | GQA |
|---|---|---|---|
| Baseline (Qwen2.5-VL-7B-Instruct) | 84.53 | 93.96 | 60.36 |
| Model Merge | 83.23 | 92.96 | 59.65 |

## C    TRAINING PARAMETERS AND TRAINING COST

All models are trained on our instruction-tuning dataset of 598K examples from MCD. We employ the AdamW optimizer with a 10% linear warm-up followed by a cosine learning rate decay. The maximum learning rate is set to $1 \times 10^{-5}$, with a batch size of 128 and a maximum sequence length of 8K tokens. Training the 8B model for two epochs takes approximately 16 hours on 8 nodes, each equipped with $8\times$A100 GPUs with 40 GB VRAMs. Training the 33B model under the same setup takes approximately 2 days.

## D    HUMAN EVALUATION

To assess the alignment between automatic evaluation and human expert judgment, we randomly sampled 100 questions from InfiBench-V and selected five MLLMs for evaluation: GPT-4o-mini, VisCodex-33B, VisCodex-8B, Qwen2.5-VL-7B-Instruct, and Qwen2.5-VL-32B-Instruct. Each model was tasked with generating responses to all sampled questions, resulting in a total of 500 model outputs.

**Annotator Details and Protocol.**    We employed three annotators, all of whom are graduate students in computer science with strong domain expertise relevant to the benchmark tasks. Although all annotators are non-native English speakers, they are proficient in English and familiar with the style and requirements of the benchmark questions.

Before the main evaluation, the annotators underwent a comprehensive training and calibration process, including practice with a subset of the dataset, review of the evaluation criteria, and targeted feedback to ensure consistent understanding of the annotation standards. A final readiness test was conducted to confirm alignment among annotators prior to the main evaluation phase.

During evaluation, all model-generated responses were anonymized and presented in a standardized format, removing all identifying information about the originating model. Annotators independently evaluated the samples without communication or influence from others, ensuring impartiality throughout the process.

For each question, the annotators were provided with the question, the model-generated answer, and the accepted StackOverflow answer as a reference. Each annotator was asked to indicate whether the model's answer correctly solved the problem (1 for correct, 0 for incorrect).

**Automatic Evaluation.**    For the automatic InfiBench-V evaluation, we adopted the same metrics as described in the main text, assigning a score in the range 0–100 to each response. For the purpose of agreement analysis with human annotations, we further mapped these scores to binary "pass" or "fail" labels according to a defined threshold. This binarization enabled direct comparison between the automatic evaluation and the binary judgments provided by human annotators.

**Agreement Analysis.**    Table 18 reports the pairwise agreement ratios (as in MT-Bench (Zheng et al., 2023)) among InfiBench-V and the human annotators, as well as between human annotators themselves. On average, InfiBench-V achieved an agreement rate of 86.07% with human annotators, which is higher than the inter-annotator agreement (79.33%). Notably, the agreement rate of InfiBench-V is comparable to that reported in previous works (Li et al., 2024c) (85.1%), demonstrating the reliability of our evaluation method.

Table 18: Agreement ratios between InfiBench-V and human annotators, as well as between annotators.

| Judge | A-1 | A-2 | A-3 | Average |
|---|---|---|---|---|
| InfiBench-V | 87.00% | 87.00% | 84.20% | 86.07% |
| A-1 | – | 80.80% | 79.20% | 80.00% |
| A-2 | 80.80% | – | 78.00% | 79.40% |
| A-3 | 79.20% | 78.00% | – | 78.60% |

**Threshold Sensitivity Analysis**    To rigorously determine the optimal mapping from continuous 0–100 scores to binary pass/fail labels, we conducted a comprehensive threshold sweep. We evaluated thresholds $\tau$ ranging from 0 to 100 with a step size of 5. For each threshold, a model response with a score $S \geq \tau$ was classified as a "pass," and the resulting labels were compared against human expert annotations to calculate the agreement rate. As shown in Table 19, the agreement rate steadily increases with the strictness of the threshold, peaking at **86.07%** for thresholds of 80 and 85. Based on this sensitivity analysis, we selected $\tau = 85$ as the decision boundary to maximize alignment with human judgment.

Table 19: Threshold sweep analysis for InfiBench-V. We evaluated agreement rates between automatic scores and human annotations across thresholds from 0 to 100. The highest agreement (86.07%) is achieved at thresholds of 80 and 85.

| Threshold | Agreement (%) | Threshold | Agreement (%) | Threshold | Agreement (%) |
|---|---|---|---|---|---|
| 0 | 68.29 | 35 | 75.54 | 70 | 85.40 |
| 5 | 68.90 | 40 | 77.85 | 75 | 85.91 |
| 10 | 68.90 | 45 | 78.39 | **80** | **86.07** |
| 15 | 69.31 | 50 | 81.98 | **85** | **86.07** |
| 20 | 69.92 | 55 | 84.01 | 90 | 85.40 |
| 25 | 73.10 | 60 | 84.62 | 95 | 83.33 |
| 30 | 73.64 | 65 | 85.40 | 100 | 81.98 |

# E    DATA STATISTICS

## E.1    DATA STATISTICS OF MCD

To ensure a fair evaluation and prevent data contamination, we performed deduplication between our dataset and the evaluation benchmarks using SimHash (Manku et al., 2007), removing any data with a similarity score greater than 0.9 to the benchmark test samples. After deduplication, the resulting dataset contains a total of 598k examples across four domains: HTML, Chart, QA, and Algorithm. Table 20 presents the statistics of the dataset, including the number of examples and the average token length (with standard deviation) for each domain.

Table 20: Statistics of the MCD by domain. For each domain, the number of samples and the average token length (mean ± standard deviation) are reported.

| Data | HTML | Chart | QA | Algorithm |
|---|---|---|---|---|
| Size | 200k | 210k | 59k | 129k |
| Avg Length | 632±144 | 551±190 | 1022±776 | 969±321 |

## E.2    DISTRIBUTION OF SELECTED BENCHMARK QUESTIONS IN INFIBENCH-V

Table 21 presents the detailed distribution of questions in InfiBench-V across various programming domains and languages. The benchmark consists of five main categories: Front-End, Back-End, Data Science & Machine Learning (DS & ML), IT Operations (IT Ops), and Mobile & Desktop development, covering a total of 13 programming languages and 322 carefully curated questions.

Table 21: Distribution of InfiBench-V questions across programming domains and languages.

| Category | Language | Count | Category Total |
|---|---|---|---|
| Front-End | CSS | 30 | |
| | HTML | 39 | 100 |
| | JavaScript | 31 | |
| Back-End | Java | 30 | |
| | PHP | 24 | |
| | Go | 5 | |
| | Ruby | 5 | 75 |
| | Rust | 6 | |
| | C++ | 5 | |
| DS & ML | Python | 90 | 95 |
| | R | 5 | |
| IT Ops | Bash | 11 | 11 |
| Mobile & Desktop | Dart | 41 | 41 |
| **Total** | | **322** | **322** |

## E.3 IMAGE QUALITY ANALYSIS OF INFIBENCH-V

To address concerns regarding the robustness of **InfiBench-V** against realistic visual noise, we conducted a quantitative analysis of image quality across all 322 benchmark samples. This analysis focuses on image resolution and clarity (measured via the Variance of Laplacian) to ensure the benchmark accurately reflects real-world scenarios.

**Resolution and Blurriness.** As shown in Table 22, the dataset retains a natural distribution of image qualities found in developer communities. Specifically, 10.56% of the samples are low-resolution ($< 50k$ pixels), typically representing small UI elements or cropped error messages. Additionally, 4.66% of the images are detected as blurry (Laplacian variance $< 100$), with extreme cases scoring as low as 1.28. This confirms that InfiBench-V includes challenging, low-quality visual inputs, testing the model's ability to reason under imperfect conditions.

**Verification of Consistency.** Despite the variation in image quality, the visual-text consistency is guaranteed by the source. We strictly filter for Stack Overflow questions with an *Accepted Answer*, ensuring that the provided images—regardless of their resolution—contain sufficient information for human experts to solve the problem.

Table 22: Image quality statistics of InfiBench-V. The presence of low-resolution and blurry images confirms that the benchmark evaluates robustness against real-world visual noise.

| Metric | Value |
|---|---|
| Total Samples | 322 |
| Average Resolution | $918 \times 552$ |
| Low Resolution ($< 50,000$ pixels) | 10.56% |
| Blurry Images (Laplacian Var $< 100$) | 4.66% |
| Extreme Blur Case (Min Variance) | 1.28 |

## E.4 CHART CATEGORY STATISTICS

We provide a statistical analysis of the chart categories contained in the Chart-to-Code portion of the MCD dataset. The chart data spans a broad range of visualization types, including common statistical plots (e.g., bar, line, pie), multi-dimensional charts (e.g., treemap, heatmap, violin), and specialized forms such as candlestick, quiver, radar, and density plots. Table 23 reports the percentage distribution across all 28 chart categories.

Table 23: Percentage distribution of chart categories within the MCD dataset (206,000 chart samples).

| Category | bar | pie | line | radar | 3d | area | combination | quiver | scatter | box |
|---|---|---|---|---|---|---|---|---|---|---|
| **Percent (%)** | 15.84 | 12.06 | 10.02 | 4.36 | 4.35 | 4.34 | 4.14 | 4.04 | 3.63 | 3.42 |
| **Category** | violin | heatmap | rose | treemap | multi-axes | bar_num | candlestick | bubble | funnel | |
| **Percent (%)** | 3.03 | 3.03 | 3.03 | 3.03 | 2.95 | 2.56 | 2.17 | 2.17 | 2.10 | |
| **Category** | ring | graph | errorbar | error_point | inset | histogram | density | | | |
| **Percent (%)** | 2.10 | 1.89 | 1.24 | 1.24 | 0.93 | 0.93 | 0.93 | | | |

# F  ANALYSIS OF MODEL MERGE

## F.1  TASK VECTOR PARAMETER-SPACE ANALYSIS

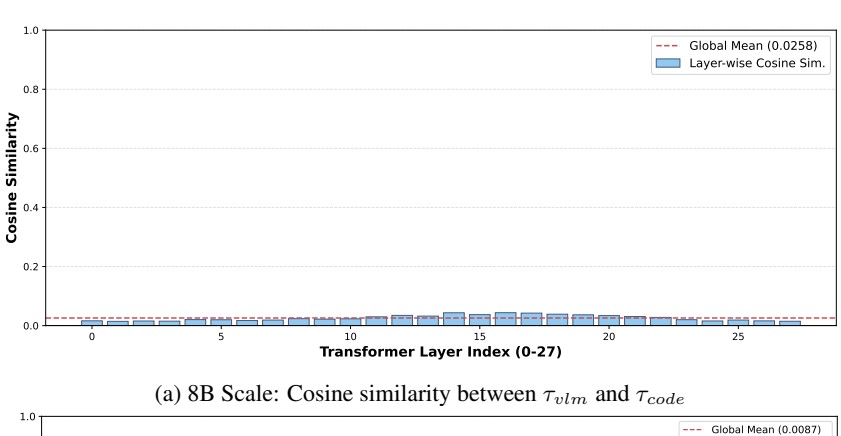

(a) 8B Scale: Cosine similarity between $\tau_{vlm}$ and $\tau_{code}$

(b) 33B Scale: Cosine similarity between $\tau_{vlm}$ and $\tau_{code}$

Figure 3: Layer-wise cosine similarity analysis of the task vectors used to construct VisCodex. (a) illustrates the orthogonality at the 7B scale (between Qwen2.5-VL-7B-Instruct and the Code LLM), and (b) illustrates the orthogonality at the 33B scale (between Qwen2.5-VL-32B-Instruct and the Code LLM). The consistently low similarity scores across all layers confirm that the parameter updates for visual grounding ($\tau_{vlm}$) and code reasoning ($\tau_{code}$) occupy disjoint subspaces, minimizing bias accumulation during merging.

To understand how the two task vectors interact during model merging, we conduct a layer-wise geometric analysis of the vision–language task vector ($\tau_{vlm}$) and the coding task vector ($\tau_{code}$). For each transformer layer $l$, we compute their cosine similarity:

$$S_l = \cos(\tau_{vlm}^l, \tau_{code}^l) = \frac{\tau_{vlm}^l \cdot \tau_{code}^l}{|\tau_{vlm}^l||\tau_{code}^l|}. \tag{5}$$

Figure 3 reports the cosine similarity for both the 8B and 33B scales. Across layers, the cosine values remain consistently small, with global means of **0.026** for the 8B model and **0.009** for the 33B model, indicating that the two task vectors exhibit only weak directional correlation in parameter space. This suggests that the update directions induced by visual–language training and code-reasoning training differ substantially across the depth of the network.

In addition to directional similarity, we also examine the layer-wise magnitudes of the task vectors. On average, $\tau_{vlm}$ has a substantially larger norm than $\tau_{code}$, with mean values of **48.31** and **18.40**, respectively, yielding an average magnitude ratio of **2.69**. While vector magnitude does not directly correspond to functional importance, these measurements provide complementary geometric information about the relative strength of the updates contributed by each task.

Taken together, the low cosine similarity and the observed magnitude relationship indicate that the two task vectors influence the parameter space in distinct and non-overwriting ways. Although this geometric analysis does not fully characterize the functional interactions between the two tasks, it offers supportive evidence consistent with our empirical finding that linear merging can retain capabilities from both vision–language and code-reasoning domains.

### F.2 REPRESENTATIONAL SIMILARITY ANALYSIS

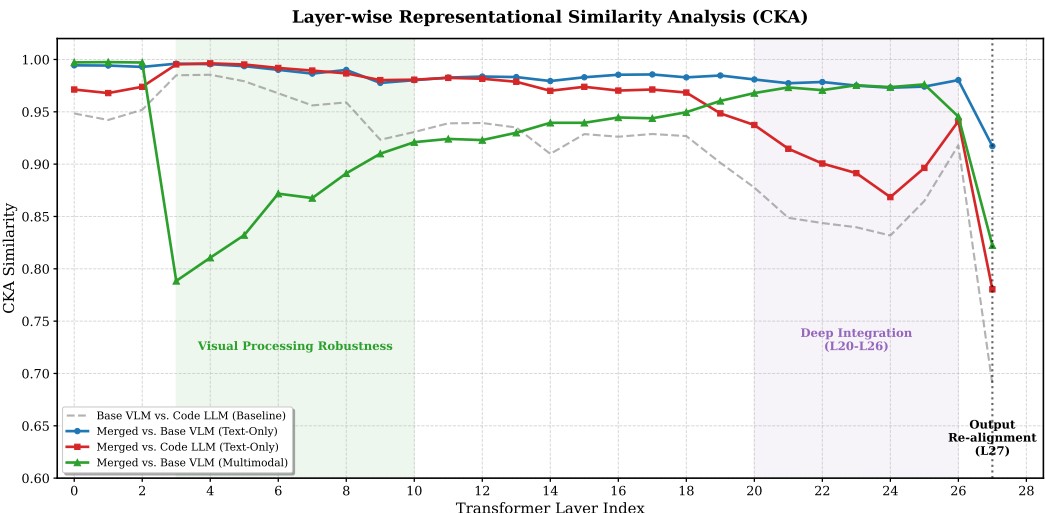

Figure 4: Layer-wise Representational Similarity Analysis (CKA). The plot illustrates three key phases: (1) Visual Processing Robustness to Parameter Shifts in early layers under multimodal inputs, (2) Deep Integration of code reasoning in layers 20–26, and (3) Output Interface Re-alignment at the final layer.

To better understand the mechanism behind our model merging strategy, we conduct a layer-wise Representational Similarity Analysis (RSA) (Kriegeskorte et al., 2008) using Centered Kernel Alignment (CKA) (Kornblith et al., 2019). We compare the internal hidden states of the **merged model** against its two source models—the Base VLM (Qwen2.5-VL-7B-Instruct) and the Code LLM (OpenCodeReasoning-Nemotron-1.1-7B). We analyze representations using input samples drawn from the MMCode (Li et al., 2024b) under two settings: (1) Text-only Inputs to probe reasoning integration, and (2) Multimodal Inputs to investigate visual processing stability. The results are illustrated in Figure 4.

**Integration of Reasoning Capabilities (Layers 20–26).** A critical question regarding model merging is how distinct capabilities are combined in deeper layers. Our analysis shows that under text-only reasoning prompts, the Base VLM and Code LLM exhibit representational divergence in deep layers, with CKA similarity dropping to the 0.83–0.91 range. In this context, the **merged model** maintains an extremely high similarity to the Base VLM ($> 0.97$) while simultaneously exhibiting increased similarity to the Code LLM ($> 0.86$) compared to the baseline. This suggests that rather than creating a distinct new "bridge" representation, the merging process effectively shifts the VLM's manifold towards the Code LLM's reasoning subspace without disrupting its original semantic continuity. This injection of code priors likely contributes to the data efficiency observed in Appendix B.6, where the **merged model** facilitates robust reasoning capabilities even in the absence of explicit algorithmic training data.

**Visual Processing Robustness to Parameter Shifts (Layers 3–10).** When processing multimodal inputs, we observe a notable trajectory in the shallow-to-middle layers (Layers 3–10), where the similarity to the Base VLM drops to approximately 0.78. This divergence likely reflects the interference introduced by the Code LLM task vector, which was trained solely on text and lacks alignment with visual tokens. Crucially, this divergence is temporary; representations successfully converge back to the VLM's semantic space in deep layers ($> 0.97$ at Layer 20+). This pattern demonstrates the intrinsic **robustness** of the VLM backbone: it effectively tolerates the parameter shifts induced by the code task vector in early layers and recovers the necessary visual semantics for final reasoning. This resilience aligns with the findings in the sensitivity analysis (Appendix B.10), where high visual performance is maintained despite these representational perturbations.

**Output Space Alignment (Layer 27).** At the final layer, the model shows a decisive alignment back towards the Base VLM distribution (similarity stays high at 0.91), while similarity to the Code LLM decreases. This behavior is mechanistically consistent with the observation in Appendix F.1, where the VLM task vector has a significantly larger magnitude than the Code task vector. The dominance of the VLM direction at the output layer ensures that the enriched internal representations are projected back into the VLM's instruction-following subspace. This alignment preserves the model's conversational interface and general multimodal capabilities (e.g., TextVQA), mitigating catastrophic forgetting as observed in Appendix B.12.

### F.3 COMPARISON WITH BACKBONE REPLACEMENT STRATEGY

To further validate the superiority of our model merging strategy over the direct backbone replacement baseline (as discussed in Table 4), we conducted a CKA analysis on the Design2Code task. This analysis probes how well the models maintain representational alignment with the original VLM when processing multimodal inputs (images + instructions). We compared the internal representations of the **Merged Model** against the **Backbone Replacement Model** (i.e., using the Code LLM backbone directly with the original vision encoder).

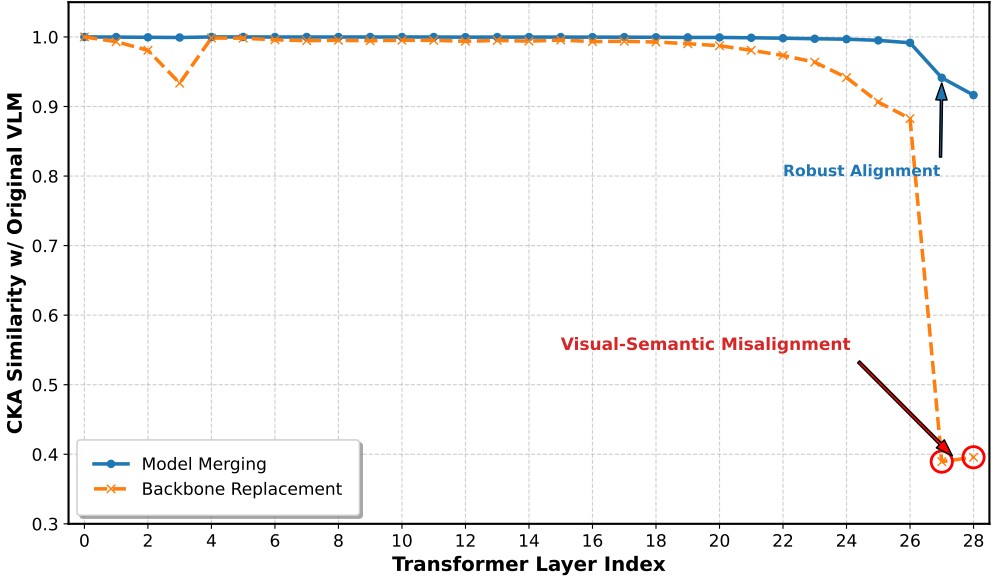

Figure 5: **CKA Analysis on Design2Code.** Comparison of representational similarity to the original VLM. The Merged Model (Blue) maintains high alignment across all layers. In contrast, the Backbone Replacement Model (Orange) exhibits a significant **Visual-Semantic Misalignment** in the final layers (CKA drops to $\sim 0.39$). This divergence indicates that without the initialization provided by model merging, the direct backbone replacement fails to preserve the critical alignment between the visual encoder and the language reasoning space.

The results, illustrated in Figure 5, reveal a critical representational divergence in the backbone replacement strategy:

- **Visual Semantic Alignment (Merged Model):** The Merged Model maintains consistently high CKA scores ($> 0.91$) across all transformer layers, achieving an overall average CKA of **0.994**. This confirms that our merging process preserves the effective alignment between the pre-trained vision projector and the language backbone. Consequently, the model effectively interprets visual tokens and maps them to the appropriate semantic space without requiring extensive re-alignment training.

- **Visual-Semantic Misalignment (Backbone Replacement):** While the Replacement Model shows high similarity in shallow layers (attributable to the shared ancestry of the base models), resulting in a high overall average CKA of **0.939**, it exhibits a **sharp divergence** in the deep layers (Layers 27–28), where CKA scores drop to $\sim 0.39$. We identify this phenomenon as **visual-semantic misalignment**: although low-level features are processed similarly, the pure Code LLM backbone—lacking the "mixed" initialization provided by merging—fails to project these visual features into the correct instruction-following subspace at the output stage.

This analysis empirically demonstrates why Model Merging serves as a superior initialization strategy. Unlike backbone replacement, which introduces a mismatch in visual-textual alignment and necessitates substantial computational resources to re-align the projector, Model Merging seamlessly grafts code capabilities while keeping the multimodal interface intact, enabling more efficient and robust fine-tuning.

# G PROMPT

## G.1 PROMPT TEMPLATE FOR GPT-4O AUTOMATIC EVALUATION ON INFIBENCH-V

---

### Prompt Template for GPT-4o Automatic Evaluation on InfiBench-V

You are a professional code assistant evaluation expert. Your task is to assess the quality of a model-generated answer to a programming-related question by comparing it with the gold reference answer. The question includes both text and a screenshot image (which may contain code, error messages, or UI context). You must consider both modalities when evaluating the answer.

Please first read the **question (text + image)**, then read the **model-generated answer**, and compare it carefully with the **reference (gold) answer**.

You must provide clear and detailed justifications before assigning scores. The scoring dimensions are weighted as follows: Correctness (50 points) and Completeness (50 points).

Below is a programming question-answering example.

**[Question]** {question}

**[Note]** The screenshot image provided alongside this question is part of the question context. You must use it to fully understand the problem being asked.
**[Screenshot Image]**
{base64_image}

**[Model Answer]**
{model_answer}

**[Reference Answer]**
{reference_answer}

Please evaluate the model's answer based on the following two dimensions:

1. **Correctness (0–50)**: Is the answer factually accurate and technically correct? Does it address the problem without errors or misleading information?
2. **Completeness (0–50)**: Does the answer cover all critical elements present in the reference answer? Are any key details missing?

—

**Justification:**
1. Correctness: <Your reasoning here>
2. Completeness: <Your reasoning here>

**Scores:**

- Correctness: X/50
- Completeness: X/50

**Total Score: X/100**

---

### G.2 PROMPT TEMPLATE FOR IMAGE-DRIVEN HTML GENERATION

**Prompt Template for Image-Driven HTML Generation**

I will provide you with an HTML image. Please design a perfect HTML based on the style of this image.

Include all CSS code in the HTML file itself. Do not assume or include any dependencies on external files. There is no need to incorporate JavaScript for dynamic interactions. Ensure careful attention to details such as element sizes, text formatting, positioning, colors, and the overall layout. Ensure all content is fully visible within a single, non-scrollable screen, and delete or redesign any parts that exceed this boundary. If the original style does not meet your needs, feel free to change it.

{HTML Image}

Your response should include the complete content of the HTML and CSS file:

# H   ADDITIONAL EXAMPLES

## H.1   MMCODE CASE 1

**Question:** Alice has a binary string. She thinks a binary string is beautiful if and only if it doesn't contain the substring "010".
In one step, Alice can change a 0 to a 1 or vice versa. Count and print the minimum number of steps needed to make Alice see the string as beautiful.
**Example**
$b = $ **010**
She can change any one element and have a beautiful string.
**Sample Input 0**
```
 STDIN     Function
 -----     --------
   7          length of string n = 7
 0101010    b = '0101010'
```
**Sample Output 0**
2
**Explanation 0:**
In this sample, $b = $ "0101010"
The figure on the right shows a way to get rid of each instance of "010": Make the string beautiful by changing 2 characters ($b[2]$ and $b[5]$).

**Qwen2.5-VL-7B-Instruct:**

```python
def beautifulBinaryString(b):
    b = list(b)
    count = 0
    for i in range(len(b) - 2):
        if b[i] == '0' and b[i + 1] == '1' and b[i + 2] == '0':
            b[i] = '1'
            b[i + 1] = '0'
            count += 2
    return count
```

✗

**VisCodex-8B:**

To solve the problem of making a binary string "beautiful" (i.e., removing all occurrences of the substring "010"), we need to identify and count these substrings within the given binary string. For each occurrence of "010", we will determine the minimal changes required to eliminate it.
Here's how we can approach the problem:
1. **Identify Occurrences**: Traverse through the string and find all instances of the substring "010".
2. **Count Changes**: For each found "010", determine the minimal number of changes required to break this pattern. Typically, changing either the middle '1' to '0' or the last '0' to '1' suffices.
3. **Sum Up Changes**: Sum up the changes for all identified "010" substrings to get the total number of steps required.

Let's implement this logic in Python:

```python
def beautifulBinaryString(b):
    # Initialize the counter for the number of changes
    changes = 0

    # Iterate over the string to find all occurrences of "010"
    i = 0
    while i <= len(b) - 3:
        if b[i:i+3] == "010":
            # We found a "010" pattern, increment the changes counter
            changes += 1
            # Move the index past this pattern
            i += 3
        else:
            # Move to the next character
            i += 1

    return changes
```

✓

Figure 6: Examples of MMCode problems that are successfully solved by VisCodex-8B but not by Qwen2.5-VL-7B-Instruct.

## H.2   MMCODE CASE 2

**Question:** You have most definitely heard the legend of King Arthur and the Knights of the Round Table. Almost all versions of this story proudly point out that the roundness of the Round Table is closely related to Arthur's belief of equality among the Knights. That is a lie! In fact, Arthur's choice of table is conditioned by his childhood traumas.

In fact, Arthur was forced to clean up quadratic tables from a young age after a tournament in pick-up sticks[1] had been played on them. After the tournament, typically there would be a bunch of sticks on the table that do not touch each other. In the spirit of the game, the organizers issued strict regulations for the table cleaners. More precisely, the sticks on the table need to be removed one by one in a way that the cleaners pull them in the shortest way towards the edge of the table closest to where they are currently sitting. They also mustn't rotate or touch the other sticks while doing this (not even in the edge points).

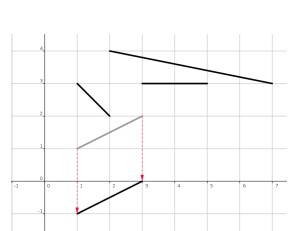

In this task, we will represent the table in the coordinate system with a square that has opposite points in the coordinates $(0, 0)$ and $(10\,000, 10\,000)$, whereas the sticks will be represented with straight line segments that lie within that square. We will assume that Arthur is sitting at the edge of the table lying on the $x$-axis. Then the movement of the stick comes down to translating the line segment along the shortest path towards the $x$-axis until the stick falls off the table (as shown in the right image). It is your task to help Arthur determine the order of stick movements that meets the requirements from the previous paragraph.

**Qwen2.5-VL-7B-Instruct**:

```python
import sys
input = sys.stdin.readline

def solve():
    n = int(input())
    ls = []
    for _ in range(n):
        (x1, y1, x2, y2) = map(int, input().split())
        ls.append((x1, y1, x2, y2))
    ls.sort(key=lambda x: (x[1] - x[3], x[0]))
    ans = []
    for (x1, y1, x2, y2) in ls:
        if y1 == y2:
            ans.append(1)
        else:
            ans.append(2)
    print(*ans)
solve()
```

✗

**VisCodex-8B**:

```python
def remove_sticks_order(n, sticks):
    # Create a list of tuples where each tuple is (index, x1, y1, x2, y2)
    indexed_sticks = [(i + 1, x1, y1, x2, y2) for i, (x1, y1, x2, y2) in enumerate(sticks)]

    # Sort the sticks based on the y-coordinate in descending order
    indexed_sticks.sort(key=lambda x: (-x[2], -x[4]))

    # Extract the indices of the sorted sticks
    sorted_indices = [stick[0] for stick in indexed_sticks]

    return sorted_indices

# Example usage:
if __name__ == "__main__":
    import sys
    input = sys.stdin.read
    data = input().split()

    n = int(data[0])
    sticks = []
    index = 1
    for _ in range(n):
        x1 = int(data[index])
        y1 = int(data[index + 1])
        x2 = int(data[index + 2])
        y2 = int(data[index + 3])
        sticks.append((x1, y1, x2, y2))
        index += 4

    result = remove_sticks_order(n, sticks)
    print(" ".join(map(str, result)))
```

✓

Figure 7: Examples of MMCode problems that are successfully solved by VisCodex-8B but not by Qwen2.5-VL-7B-Instruct.

## H.3 InfiBench-V Case 1

**Question:** I have a dataset with absolute numbers for various variants over time, and I want to create a plot similar to the one shown in the image, depicting the abundance of these variants over time. I am working in Python, and I would prefer to use matplotlib, although other libraries are also acceptable if necessary.
Does Python offer any specific function or tool for generating this type of plot? Additionally, what is the common name for this kind of plot, as I have encountered it before but am having trouble finding relevant resources online?

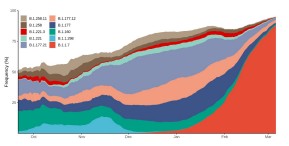

**Response**: The type of plot shown in the image is commonly known as an "area plot" or "stacked area plot." In Python, you can create such plots using the `matplotlib` library. Although there isn't a specific function called "area plot" in `matplotlib`, you can use the `stackplot` function to generate this type of visualization.
Here's a basic example of how you can create a stacked area plot using `matplotlib`:

```python
import matplotlib.pyplot as plt
import numpy as np

# Example data
dates = np.arange('2021-01', '2021-03', dtype='datetime64[D]')
variant1 = np.random.rand(len(dates))
variant2 = np.random.rand(len(dates))
variant3 = np.random.rand(len(dates))

# Normalize data to sum to 1 for each day
total = variant1 + variant2 + variant3
variant1 /= total
variant2 /= total
variant3 /= total

# Create stacked area plot
plt.stackplot(dates, variant1, variant2, variant3, labels=['Variant 1', 'Variant 2', 'Variant 3'], alpha=0.8)
plt.legend(loc='upper left')
plt.xlabel('Date')
plt.ylabel('Frequency (%)')
plt.title('Abundance of Variants Over Time')
plt.show()
```

In this example, `variant1`, `variant2`, and `variant3` are arrays representing the frequencies of different variants over time. You should replace these with your actual data. Make sure your data is normalized (i.e., the sum of all variants for each time point should equal 1, or 100% if expressed in percentage terms) before plotting.
If you prefer using `pandas`, you can also leverage its capabilities to simplify data manipulation and plotting:

```python
import pandas as pd
import matplotlib.pyplot as plt

# Example data
data = {
    'Date': pd.date_range(start='2021-01-01', periods=60, freq='D'),
    'Variant1': np.random.rand(60),
    'Variant2': np.random.rand(60),
    'Variant3': np.random.rand(60)
}

df = pd.DataFrame(data)
df.set_index('Date', inplace=True)

# Normalize data to sum to 1 for each day
df = df.div(df.sum(axis=1), axis=0)

# Plot
df.plot.area(alpha=0.8)
plt.ylabel('Frequency (%)')
plt.title('Abundance of Variants Over Time')
plt.show()
```

This approach simplifies the handling of data, especially if it's already in a `pandas` DataFrame format. Adjust the data and labels as needed for your specific dataset.

**Category: DS & ML——Python**
**Evaluation Criteria: Keyword Matching & GPT-4o**
**Keywords:** `stackplot`, `matplotlib`
**Keyword Matching Score:** 100
**GPT-4o Score:** 100

**Total Score: 100**

Figure 8: A case of InfiBench-V from the DS & ML category.

## H.4  INFIBENCH-V CASE 2

**Question:** I attempted to change the colors in my macOS Terminal by following a guide and modifying my `.bash_profile` file. After creating and saving the `.bash_profile`, every time I open the Terminal, I encounter the following errors as shown in the screenshot on the right:
Despite deleting the `.bash_profile`, the issue persists. Every command I enter is followed by these errors:

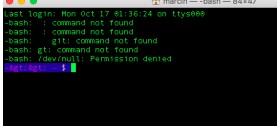

```
-bash: git: command not found
-bash: gt: command not found
-bash: /dev/null: Permission denied
```

Here is the content I added to the `.bash_profile` file:

```
export CLICOLOR=1
export LSCOLORS=GxBxCxDxexegedabagaced
parse_git_branch() {
    git branch 2> /dev/null |
    sed -e '/^[^*]/d' -e 's/* (.*)/ (\1)/'
}
export PS1="\e[0;35m->>
    \e[1;34m\W\e[0;32m\$(parse_git_branch)\e[0;37m $ "
```

Please help me resolve these issues and understand why these errors are occurring.

---

**Response**:
The errors you're encountering are likely due to incorrect syntax in your `.bash_profile` file. Let's address each issue:

1. **Incorrect Redirection Symbol**: In your `parse_git_branch()` function, you used `>` instead of `>` for redirecting output to `/dev/null`. The correct line should be:

   ```
   git branch 2> /dev/null |
   ```

2. **Extra Spaces in PS1**: Your `PS1` variable has some extra spaces that might cause issues. It should look like this:

   ```
   export PS1="\e[0;35m>> \e[1;34m\W\e[0;32m\$(parse_git_branch)\e[0;37m $ "
   ```

   Make sure there are no unintended spaces around special characters or within the escape sequences.
3. **Check for Hidden Characters**: Sometimes, copy-pasting from certain editors or websites can introduce hidden characters. Ensure your `.bash_profile` doesn't contain any non-printable characters. You can use a text editor that shows whitespace or use a command-line tool to check for them.
4. **File Permissions**: Ensure your `.bash_profile` file has the correct permissions. You can set it with:

   ```
   chmod 644 ~/.bash_profile
   ```

5. **Recreate .bash_profile**: If the problem persists, try recreating the `.bash_profile` file manually without copying and pasting. Open a new terminal and type the commands directly into a new file.

After making these changes, save the file and restart your terminal session. This should resolve the errors you're seeing. If the problem continues, double-check for any other syntax errors or unexpected characters in the file.

**Category: IT Ops——Bash**
**Judge: GPT-4o**

1. **Correctness:**
   - The model answer correctly identifies the primary issue with the `.bash_profile` file, which is the use of HTML entities (`>`) instead of the correct characters (`>`). This aligns with the reference answer.
   - The model also addresses the potential for non-printable characters and file permissions, which, while not mentioned explicitly in the reference answer, are reasonable considerations given the symptoms described.
   - Additionally, the model suggests recreating the `.bash_profile` file to ensure there are no hidden characters, which is a practical troubleshooting step that could help resolve the persistent issues.
   - Overall, the model's suggestions are factually accurate and technically correct, directly addressing the syntax errors.

2. **Completeness:**
   - The reference answer is concise and focuses solely on the issue of HTML entities, while the model answer provides a more comprehensive approach by including additional troubleshooting steps (checking for hidden characters, file permissions, and recreating the file).
   - Although the model answer includes extra steps beyond the reference answer, it does not miss any critical elements present in the reference. Instead, it enhances the response with additional useful advice for resolving the issue thoroughly.
   - The model's completeness is therefore high, as it covers the key issue identified in the reference answer and adds further depth.

**Scores:**
- Correctness: 50/50
- Completeness: 50/50

**Total Score: 100/100**

Figure 9: A case of InfiBench-V from the IT Ops category.

## H.5 INFIBENCH-V CASE 3

**Question:** I am working on a 2D game in JavaScript and I am encountering a trigonometry issue. The game involves a player shooting at a target, but there is an obstacle blocking the shot between the player and the target. I need to find the coordinates of this obstacle. Here is the information I have:

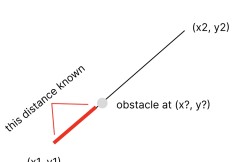

1. The starting coordinates of the shot are $(x_1, y_1)$.

2. The ending coordinates of the shot are $(x_2, y_2)$.

3. The distance from the starting coordinates to the obstacle is known.

How can I calculate the $(x, y)$ coordinates of the obstacle along the line of the shot? Additionally, I would appreciate an efficient solution implemented in JavaScript.
Please refer to the attached image for a visual representation of the problem.

**Response:**
To solve the problem of finding the coordinates of the obstacle along the line of the shot, you can use vector mathematics. Given the starting point $(x1, y1)$, the ending point $(x2, y2)$, and the distance $d$ from the starting point to the obstacle, you can calculate the coordinates of the obstacle as follows:

1. Calculate the direction vector from $(x_1, y_1)$ to $(x_2, y_2)$:

$$\text{direction} = (x_2 - x_1, \ y_2 - y_1)$$

2. Normalize the direction vector to get the unit vector in the direction of the shot:

$$\text{magnitude} = \sqrt{(x_2 - x_1)^2 + (y_2 - y_1)^2}$$

$$\text{unit\_direction} = \left( \frac{x_2 - x_1}{\text{magnitude}}, \ \frac{y_2 - y_1}{\text{magnitude}} \right)$$

3. Multiply the unit direction vector by the distance $d$ to find the displacement vector from the starting point to the obstacle:

$$\text{displacement} = (d \times \text{unit\_direction}_x, \ d \times \text{unit\_direction}_y)$$

4. Add the displacement vector to the starting point to get the coordinates of the obstacle:

$$\text{obstacle}_x = x_1 + \text{displacement}_x$$

$$\text{obstacle}_y = y_1 + \text{displacement}_y$$

Here's how you can implement this in JavaScript:

This code will give you the coordinates of the obstacle along the line of the shot, given the specified distance from the starting point.

**Category: Front-End——JavaScript**
**Evaluation Criteria: Unit Testing**
**Sample Test Case:**
```
input:  {"x1":0.0,"y1":0.0,"x2":10.0,"y2":0.0,"distance":5.0},
expected:  [5.0, 0.0],
actual:  [5.0, 0.0], ...
```
(only partial cases shown)
**Total Score: 100**

Figure 10: A case of InfiBench-V from the Front-End category.

## H.6 INFIBENCH-V CASE 4

**Question:** I am developing a Flutter application where I want to draw a square on the screen at the location of a click. The square should be centered at the click point, but currently, it is being drawn at seemingly random positions. I am using 'event.pointerEvent.localPosition.dx' and 'event.pointerEvent.localPosition.dy' to obtain the click coordinates, and I am passing these coordinates to 'Rect.fromCenter' to draw the square.
Below is my Flutter code, which includes the 'SpriteSquare' widget and the 'handleAdd' function from its parent widget, 'MySpriteGame'.

**sprite_square.dart**

```dart
class SpriteSquare extends NodeWithSize {

  Color color;
  Function handler;
  double left;
  double top;

  SpriteSquare(size, this.color, this.handler, this.top, this.left) :
      super(size){
    userInteractionEnabled = true;
  }

  @override handleEvent(SpriteBoxEvent event) {
    handler(event);
    return true;
  }

  @override
  Future<void> paint(Canvas canvas) async {
    canvas.drawRect(
      Rect.fromLTWH(left, top, size.height, size.width),
      Paint()..color = color
    );
  }
}
```

**handleAdd**

```dart
handleAdd(SpriteBoxEvent event) {
    final _whitesquare = SpriteSquare(
      const Size(50.0, 50.0),
      const Color(0xFFFFFFFF),
      handleRemoveSelf,
      Offset(event.pointerEvent.localPosition.dx,
          event.pointerEvent.localPosition.dy)
    );
    _background.addChild(_whitesquare);
    return true;
}
```

Can you help me understand why the square is not being drawn centered at the click location and suggest how to fix this issue?

**Response**:
The issue you're experiencing with the square not being drawn centered at the click location is likely due to the way you're calculating the position of the square. In your current implementation, you're using 'Rect.fromLTWH' to draw the rectangle, which takes the left and top coordinates as the starting point of the rectangle. This means that the rectangle is drawn starting from the top-left corner at the specified coordinates, rather than being centered at those coordinates.
To center the square at the click location, you need to adjust the 'left' and 'top' values by subtracting half of the square's width and height, respectively. Here's how you can modify your code:
In the 'handleAdd' function, calculate the 'left' and 'top' values like this:

```dart
handleAdd(SpriteBoxEvent event) {
    final clickX = event.pointerEvent.localPosition.dx;
    final clickY = event.pointerEvent.localPosition.dy;

    // Calculate the top-left corner of the square so that it is centered at the click location
    final left = clickX - 25.0; // Half of the square's width (50.0 / 2)
    final top = clickY - 25.0; // Half of the square's height (50.0 / 2)

    final _whitesquare = SpriteSquare(
      const Size(50.0, 50.0),
      const Color(0xFFFFFFFF),
      handleRemoveSelf,
      left,
      top
    );
    _background.addChild(_whitesquare);
    return true;
}
```

By adjusting the 'left' and 'top' values in this way, the square will be centered at the click location. This is because you're effectively moving the top-left corner of the square to a position that centers the square around the click point.

**Category: Mobile & Desktop——Dart**
**Evaluation Criteria: GPT-4o**
**Correctness Score: 35/50**
**Completeness Score: 25/50**
**Total Score: 60**

Figure 11: A case of InfiBench-V from the Mobile & Desktop category.

## H.7 INFIBENCH-V CASE 5

**Question:** In the context of a Ruby on Rails application using version 2.3.5 and Ruby 1.8.7, I am working with three database tables: A, B, and C. The relationships are as follows: table A has multiple associated records in table B, and table C also has multiple associated records in table B. There is no direct relationship between tables A and C.

I want to query records from table A and include associated records from tables B and C. I can include table B in the query using the following code:

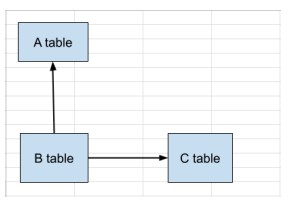

```
A.all(:conditions => { :id => id }, :include => [:B])
```

However, I am unsure how to also include table C in the query. Despite extensive research, I have not found a solution. How can I modify the query to include records from both tables B and C when querying table A's records?

**Response**:

To include associated records from both tables B and C when querying table A in a Ruby on Rails application, you need to ensure that the associations are set up correctly in your models. Assuming you have the following associations:
- Table A has many records in table B (e.g., `has_many :bs`).
- Table C has many records in table B (e.g., `has_many :cs`).
You can use eager loading with `:include` to load associated records from both tables B and C. Here's how you can modify your query:

```
A.all(:conditions => { :id => id }, :include => { :bs => :cs })
```

This assumes that:
- In model A, you have `has_many :bs`.
- In model B, you have `belongs_to :a` and `has_many :cs`.
- In model C, you have `has_many :bs`.
If the association between B and C is different, adjust the `:include` statement accordingly. If there is no direct relationship between A and C, but you want to include C through B, make sure the association is correctly defined in your models.

**Note:** Since you're using Rails 2.3.5 and Ruby 1.8.7, the syntax for eager loading is slightly different compared to newer versions of Rails. Make sure your models' associations are set up correctly to reflect these relationships.

**Category: Back-End——Ruby**

**Evaluation Criteria: GPT-4o**

1. **Correctness (0–50):** The model answer provides a solution using Rails' eager loading feature with the `:include` option in a way that aligns with Rails 2.3.5 syntax. It correctly assumes that B acts as an intermediary between A and C. The proposed query modification `A.all(:conditions => { :id => id }, :include => { :bs => :cs })` is a valid way to include associated records from both tables B and C through B. This matches the associations depicted in the screenshot, where A is linked to B, and B is linked to C. The model answer elaborates on the associations needed in the models, which is technically correct. However, the reference answer is more concise and directly points to the usage of `:include => { :B => :C }` without additional assumptions about model setup. The model answer's explanation about associations, while correct, is not necessary to resolve the specific query issue. Nonetheless, the model's proposed query would work given the correct associations. Therefore, the correctness of the model's answer is high, albeit slightly verbose.

2. **Completeness (0–50):** The model answer provides a comprehensive response by explaining the underlying assumptions about the model associations, which are essential to achieving the desired query behavior. It goes beyond the reference answer by elaborating on how associations should be set up in the models. This additional detail provides a complete understanding of how the query interacts with the model structure. While the reference answer is more succinct, the model answer's thoroughness in explaining potential association configurations adds value and ensures that the user understands the prerequisites for the query to function as intended. Thus, the model answer is complete but includes information that, although helpful, is not strictly necessary to address the question directly.

**Scores:**
- Correctness: 45/50
- Completeness: 45/50
**Total Score: 90**

Figure 12: A case of InfiBench-V from the Back-End category.

## H.8 FAILURE CASE

Despite achieving state-of-the-art performance, VisCodex exhibits limitations when handling highly complex 3D spatial relationships or information-dense UI elements. As illustrated in Figure 13, we identify primary failure modes in **3D spatial reconstruction** (Row 1) and **fine-grained UI content generation** (Row 4), where the model struggles to precisely reconstruct coordinate connectivity or render detailed inner content.

Crucially, we observe that the backbone model, Qwen2.5-VL-7B-Instruct, exhibits similar failure patterns on these challenging samples. Since VisCodex-8B is built upon the Qwen2.5-VL-7B-Instruct architecture, it inherently inherits the visual perception capabilities—and effectively the limitations—of this foundation model. This implies that the current visual understanding bottleneck stems primarily from the base model rather than the merging strategy itself. Consequently, we anticipate that **future improvements in the visual capabilities of base models will directly drive further enhancements** in our framework's multimodal code generation performance.

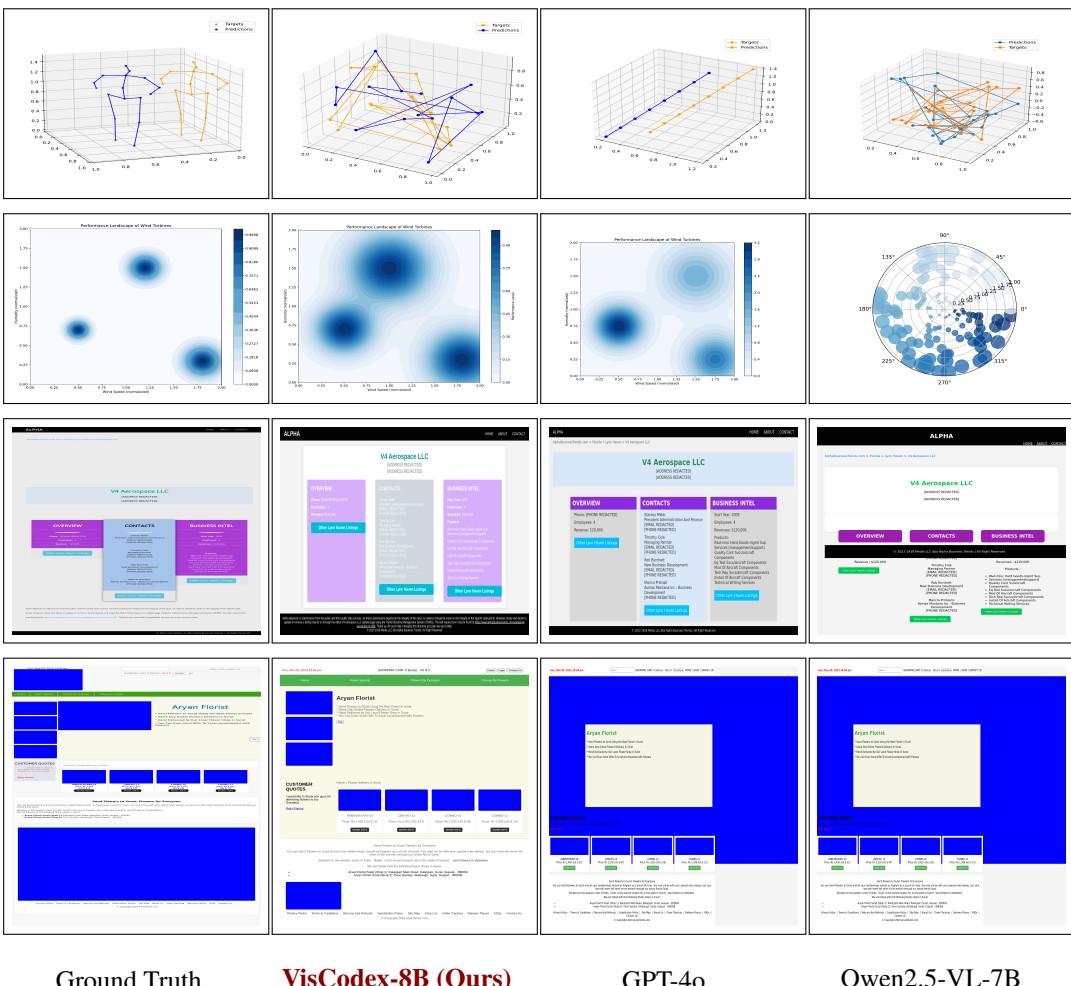

| Ground Truth | **VisCodex-8B (Ours)** | GPT-4o | Qwen2.5-VL-7B |

Figure 13: Failure case analysis on ChartMimic and Design2Code. The model struggles with 3D spatial structure and fine-grained UI details. Notably, these perceptual limitations mirror those of the backbone model, Qwen2.5-VL-7B-Instruct, indicating that VisCodex-8B's visual understanding is bounded by its foundational architecture. Future advancements in base visual models are expected to mitigate these issues.

# I    TRAINING STABILITY AND THEORETICAL FOUNDATIONS OF TASK-VECTOR MERGING

We analyze both the empirical optimization behavior and the theoretical underpinnings of task vector-based model merging to demonstrate that VisCodex maintains stable training dynamics and operates within a theoretically sound merging regime.

## I.1    TRAINING STABILITY AND LOSS DYNAMICS

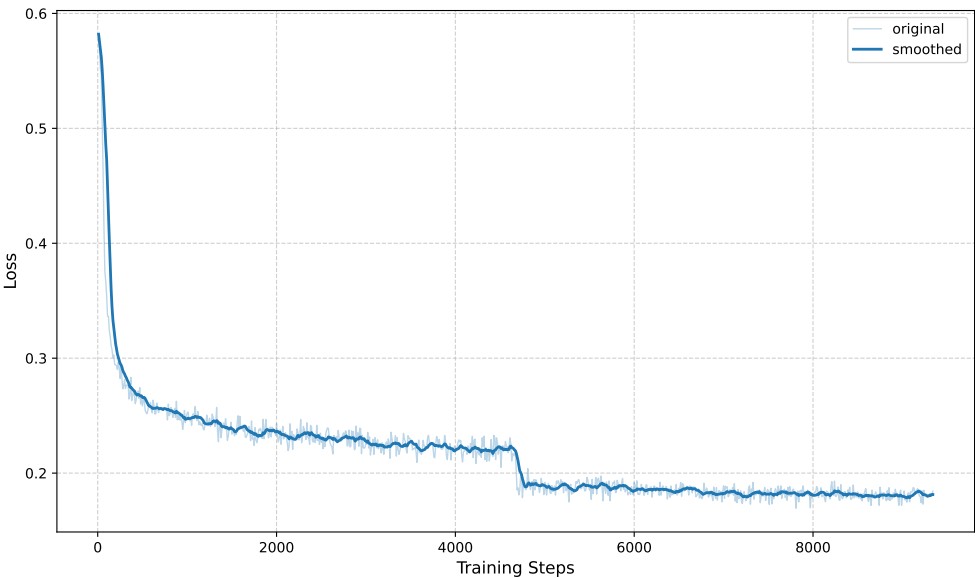

(a) Training loss of VisCodex-8B during SFT.

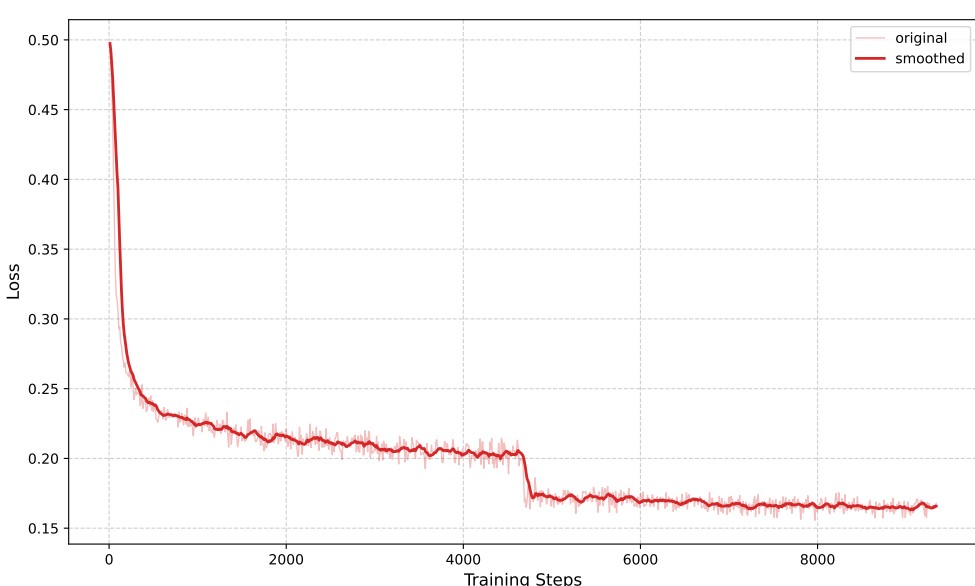

(b) Training loss of VisCodex-33B during SFT.

Figure 14: Smoothed training loss curves for VisCodex-8B and VisCodex-33B. Both models show **stable, monotonic loss reduction** throughout SFT, indicating that task-vector merging does not introduce gradient conflict or destabilize optimization.

We examine the optimization dynamics of VisCodex-8B and VisCodex-33B during supervised fine-tuning (SFT), following the one-shot task-vector merge.

As shown in Figure 14, training proceeds smoothly for both model sizes, with no spikes, oscillations, or irregularities. This confirms that the merged initialization forms a **stable starting point** for downstream training and does not cause accumulated bias or interfering gradients.

### I.2 THEORETICAL FOUNDATIONS OF TASK-VECTOR COMPOSITION

Our empirical findings align with established theoretical results on model merging and task arithmetic.

**Linear composition with minimal interference.** Task vectors encode low-curvature directions in parameter space corresponding to task-specific transformations. Prior work shows they can be linearly combined to transfer capabilities across domains without retraining and with limited interference (Ilharco et al., 2022). Fisher-weighted and interference-aware analyses further demonstrate that successful merging occurs when tasks modify disjoint or weakly overlapping parameter subsets (Matena & Raffel, 2022; Yadav et al., 2023).

Our case satisfies this condition:

- near-orthogonality between $\tau_{\text{vlm}}$ and $\tau_{\text{code}}$,
- smooth $\lambda$-sensitivity curves (Appendix B.10),
- high deep-layer CKA similarity $> 0.97$ after merging (Appendix F.2).

These observations indicate that VisCodex lies in the theoretical regime where task-vector composition is expected to succeed.

**Cross-domain ability fusion.** Recent studies show that merging supports the combination of heterogeneous skills— such as vision + mathematical reasoning (Chen et al., 2025) or textual preference integration into multimodal models (Li et al., 2025a)—because the underlying capabilities tend to occupy separated parameter subspaces. Our results extend this direction: vision–language understanding and code reasoning also exhibit such structural compatibility, enabling reliable fusion via linear task-vector arithmetic.

