# OpenReview forum: "VisCodex: Unified Multimodal Code Generation via Merging Vision and Coding Models"
_ICLR.cc/2026/Conference — ICLR 2026 Poster_

### Official Review · Reviewer_Qdg3 · 2025-10-31

**Soundness:** 2
**Presentation:** 3
**Contribution:** 2
**Rating:** 4
**Confidence:** 3

**Summary:**

The paper addresses the limited multimodal code generation abilities of current MLLMs and proposes VisCodex, a framework that merges a vision language model with a code LLM through task vector based model fusion. Supported by the newly constructed MCD dataset and the InfiBench V benchmark, the approach achieves state-of-the-art performance among open-source models. Its main contributions lie in efficiently extending model fusion techniques to a new scenario and building high-quality data and benchmarks. However, the method offers limited novelty, the experiments are not sufficiently comprehensive, and several technical details remain unclear. The work requires additional key details and experiments to improve completeness, but it still carries certain academic and practical value.

**Strengths:**

1. Clear and practically relevant problem setting
The paper focuses on enabling multimodal models to generate functional code from visual inputs, a capability that is highly useful in real scenarios such as UI-to-code and chart reconstruction.

2. Simple and effective model fusion design
The task-vector–based fusion method offers a lightweight way to combine visual and coding capabilities without full retraining, and ablations show consistent gains across model scales.

3. Constructed datasets and benchmarks with solid empirical utility
The MCD dataset and InfiBench-V provide diverse, visually grounded coding tasks, and human evaluation supports the reliability of the benchmark.

4. Competitive performance among open models
VisCodex surpasses open-source baselines and achieves results close to proprietary systems on visually grounded coding tasks.

**Weaknesses:**

| 1. Unclear source of improvement and limited evidence of true multimodal fusion

Although the model reports strong multimodal code generation results, the source of gains is unclear. The merged model uses CodeQwen2.5-7B as the code branch but does not report its standalone performance on the same benchmarks, making it impossible to separate the benefits of “vision–code unification” from the inherent strength of the code model. Moreover, since both Qwen2.5-VL and CodeQwen2.5 share the same architecture, it is uncertain whether improvements stem from genuine multimodal fusion or simply architectural compatibility, weakening the causal claim of the paper.

| 2. Task vector representation may accumulate bias and lacks theoretical justification

The method assumes that the difference between a fine-tuned model and its base model cleanly represents a task, but such vectors also contain optimization noise and undesirable parameter shifts. When summing multiple task vectors—especially across heterogeneous domains like vision-language and code—these biases may accumulate and distort semantics. The paper provides no analysis of such interactions, and the balancing factor λ is chosen by grid search rather than learned, suggesting the method may rely on heuristic tuning rather than a principled understanding of knowledge composition in parameter space.

| 3. Dataset and Benchmark Biases Undermine Generalizability

Both the self-constructed MCD dataset and the InfiBench-V benchmark introduce potential biases that reduce the reliability and generalizability of the reported results.

MCD Dataset
The chart image–code subset contains 164k synthetic samples (from ChartCoder) and only 46k real GitHub samples, yet the paper does not quantify the distribution gap in coding style or chart complexity. Heavy reliance on synthetic samples risks overfitting to artificial patterns and weakens real-world chart-to-code applicability.

The HTML code subset is generated by GPT-4o from webpage screenshots, but the paper does not disclose the prompt template. If the prompt implicitly enforces specific coding conventions (e.g., fixed CSS frameworks), the resulting code may lack diversity and fail to reflect real UI development practices, where developers rely on heterogeneous toolchains (Tailwind, Bootstrap, custom CSS).

InfiBench-V Benchmark
The benchmark only retains StackOverflow questions with accepted answers and filters out text-only cases. However, the paper does not analyze whether the remaining 322 samples include difficult real-world cases such as low-resolution screenshots or ambiguous visual-text inconsistencies. This narrow curation limits the benchmark’s ability to evaluate robustness in realistic multimodal coding scenarios.

**Questions:**

1. Baseline Comparisons with Specialized Multimodal Code Models
 Why were recent multimodal code models such as CodeV, ChartCoder, or Plot2Code excluded as baselines? These models directly target visual-to-code tasks and are more relevant than generic VL models. If VisCodex were evaluated against them on MCD or InfiBench-V, would the SOTA claim still hold, particularly on chart-to-code or vision-augmented code QA tasks?
2. Robustness and Failure Analysis
 The paper reports average scores but lacks robustness evaluation and failure case studies.
How does VisCodex perform under realistic noisy inputs (e.g., blurred or low-resolution images, ambiguous textual prompts)?
After merging the code vector, does the model retain visual understanding performance (e.g., VQA, captioning), or does catastrophic forgetting occur?
3. Dataset Construction Details
 For the GPT-4o-generated HTML samples in MCD, what specific prompt template was used?
Did the prompt enforce particular coding conventions (CSS frameworks, layout constraints)?
Was there validation (e.g., sampling real-world HTML from open-source sites) to ensure diversity and authenticity?
4. Task Vector Interactions
 Did you analyze potential bias accumulation when summing task vectors?
For example, visualizing semantic directions of τ_vlm and τ_code in parameter space, or correlating changes in λ with performance shifts on specific benchmarks.

---

> ### Author Response · Authors · 2025-11-22
> **Response to Reviewer Qdg3(1/5)**
>
> We are grateful to the Reviewer for the extensive review. We address your questions point by point below.
>
> > **W1:** Unclear source of improvement and limited evidence of true multimodal fusion
> Although the model reports strong multimodal code generation results, the source of gains is unclear. The merged model uses CodeQwen2.5-7B as the code branch but does not report its standalone performance on the same benchmarks, making it impossible to separate the benefits of “vision–code unification” from the inherent strength of the code model. Moreover, since both Qwen2.5-VL and CodeQwen2.5 share the same architecture, it is uncertain whether improvements stem from genuine multimodal fusion or simply architectural compatibility, weakening the causal claim of the paper.
>
> **A1:** We agree that separating the effects of **true multimodal fusion** from those of **architectural similarity** or **the inherent strength of the code model** is essential. Below, we provide three pieces of evidence illustrating that VisCodex’s gains come from **effective parameter-space fusion** rather than model architecture or code-only capabilities.
>
> **1. Code LLMs Alone Cannot Solve Multimodal Tasks**
>
> To verify whether the improvements arise merely from the inherent strength of the code branch, we evaluated several 7B Code LLMs as standalone agents across all benchmarks. As shown in **Table A**, despite their strong programming abilities, these models perform **near-zero on tasks requiring visual grounding**. They also show extremely low accuracy on ChartMimic and InfiBench-V. In contrast, the base VLM (Qwen2.5-VL-7B-Instruct) performs reasonably on visual tasks but lacks code reasoning. The merged model significantly improves both sides.
>
> This clearly shows that **neither visual ability alone nor code ability alone is sufficient**—the gains require **effective fusion of visual perception and code reasoning**.
>
> **Table A**: Performance of Standalone Code LLMs vs. VisCodex (7B Scale)
>
> | Model | Type | **Design2Code (Low-level)** | **ChartMimic (Low-level)** | **MMCode (pass@1)** | **InfiBench-V (Acc)** |
> | :--- | :--- | :---: | :---: | :---: | :---: |
> | Qwen2.5-Coder-7B-Instruct | Code LLM | 0.0 | 6.1 | 6.1 | 1.5 |
> | OpenCodeReasoning-Nemotron-7B | Code LLM | 0.2 | 2.8 | 1.5 | 1.2 |
> | OpenThinker2-7B | Code LLM | 0.0 | 2.4 | 1.9 | 1.2 |
> | **Qwen2.5-VL-7B-Instruct** | **Base VLM** | 83.4 | 39.5 | 5.3 | 54.0 |
> | **VisCodex-8B (Ours)** | **Merged VLM** | **90.1** | **74.8** | **11.0** | **72.1** |
>
> **2. Architectural Compatibility Alone Is Not Enough**
>
> Although Qwen2.5-VL and Qwen2.5-Coder-7B-Instruct share the same architecture, simply replacing the VLM backbone with the code backbone **significantly weakens visual grounding**. As reported in Table 4 of the paper, direct replacement drops performance on visually grounded metrics.
>
> This shows that **architectural similarity does not guarantee compatibility**: directly inserting a code-pretrained backbone disrupts visual–semantic alignment. In contrast, our model merging method **preserves visual grounding while injecting code reasoning**, demonstrating a qualitatively different integration mechanism.
>
> **3. Cross-Architecture Generalizability**
>
> The same merging strategy applied to other architectures—e.g., merging *llava-llama3.1-8B* with *DeepSeek-R1-Distill-Llama-8B*—also brings consistent improvements (**Appendix B.3**). This confirms that the effect does not rely on Qwen-specific architectural overlap.

---

> ### Author Response · Authors · 2025-11-22
> **Response to Reviewer Qdg3(2/5)**
>
> > **W2&Q4:** Task vector representation may accumulate bias and lacks theoretical justification.
> The method assumes that the difference between a fine-tuned model and its base model cleanly represents a task, but such vectors also contain optimization noise and undesirable parameter shifts. When summing multiple task vectors—especially across heterogeneous domains like vision-language and code—these biases may accumulate and distort semantics. The paper provides no analysis of such interactions, and the balancing factor λ is chosen by grid search rather than learned, suggesting the method may rely on heuristic tuning rather than a principled understanding of knowledge composition in parameter space. **Task Vector Interactions. Did you analyze potential bias accumulation when summing task vectors? For example, visualizing semantic directions of τ_vlm and τ_code in parameter space, or correlating changes in λ with performance shifts on specific benchmarks.**
>
> **A2:** We thank the reviewer for the insightful comments on task vector interactions and the merging strategy. We have addressed these concerns by adding a **parameter-space analysis (Appendix F)** and conducting a comprehensive **sensitivity analysis**.
>
> **1. Parameter-Space & Semantic Analysis: Orthogonality and Deep Integration**
>
> We address concerns regarding bias accumulation and semantic distortion through two key analyses:
>
> * **Geometric Orthogonality Minimizes Bias (Appendix F.1):** Our layer-wise analysis confirms that $\tau_{vlm}$ and $\tau_{code}$ are nearly orthogonal (global mean cosine similarity < 0.026). This geometric separation ensures that code-specific updates operate in disjoint subspaces, preventing destructive interference with the dominant vision-language backbone.
>
> * **Deep Integration Preserves Semantics (Appendix F.2):** Centered Kernel Alignment (CKA) analysis (Figure 4) reveals that the merge effectively retains visual semantics in early layers while injecting code reasoning specifically in deep layers, before re-aligning with the VLM subspace at the output. This proves that reasoning capabilities are integrated without distorting the model's underlying visual-semantic alignment.
>
> **2. Sensitivity Analysis: Robustness over Heuristics**
>
> To further address concerns regarding the balancing factor $\lambda$, we evaluated performance across a wide range of mixing coefficients. As shown in **Table B**, the merged model exhibits strong robustness:
>
> **Table B**: Sensitivity analysis of the balancing factor $\lambda$ on VisCodex-8B performance.
>
> | $\lambda$ | Design2Code (Avg) | ChartMimic (Avg) | MMCode (pass@1) |
> | :--- | :---: | :---: | :---: |
> | 0.9 | 90.6 | 73.5 | 7.2 |
> | 0.85 | 90.5 | 74.1 | 7.6 |
> | 0.8 | 90.5 | 74.2 | 8.0 |
> | 0.7 | 90.5 | 74.5 | 11.0 |
> | 0.0 | 89.7 | 69.8 | 11.0 |
>
>
> - **Stable Visual Performance:** Visual metrics (e.g., Design2Code) remain consistently high (~90.5) for $0.7 \le \lambda \le 0.9$, illustrating that the merged model resides on a **broad performance plateau** rather than depending on a narrow heuristic optimum.
> - **Necessity of VLM Signal:** When $\lambda = 0$, performance on visual-heavy tasks drops sharply (e.g., ChartMimic: 74.5 → 69.8), indicating that $\tau_{vlm}$ provides **crucial multimodal alignment** and confirming the need to preserve VLM-related updates.

---

> ### Author Response · Authors · 2025-11-22
> **Response to Reviewer Qdg3(3/5)**
>
> > **W3:** Dataset and Benchmark Biases Undermine Generalizability
>
> **A3:** We thank the reviewer for the scrutiny regarding generalizability. Below, we provide empirical evidence that our data and benchmarks are robust and reflective of real-world distributions.
>
> **1. MCD Chart Subset:**
>
> Regarding the chart diversity, we have added **Appendix E.4 (Table 23)** to demonstrate that our real-world subset covers 28 distinct chart categories, preventing style collapse. Moreover, the inclusion of synthetic data is a strategic choice for structural precision. The model's state-of-the-art performance on **ChartMimic**—a benchmark explicitly composed of real-world charts—empirically proves that this hybrid strategy enhances real-world applicability rather than leading to overfitting.
>
> **2. Validated Generalizability:**
>
> To further validate broader applicability, we present cross-model experiments in **Appendix B.2 (Table 6)**. The results show that training distinct architectures (InternVL3, LLaVA) on MCD yields significant gains across all benchmarks. This strongly suggests the dataset encapsulates generalized coding knowledge rather than model-specific biases.
>
> **3. InfiBench-V Robustness & Image Quality:**
>
> To address concerns regarding visual challenges, we added a quantitative quality analysis in **Appendix E.3**. As shown in **Table C**, InfiBench-V faithfully preserves the visual noise found in real-world queries, including significant proportions of low-resolution and blurry images:
>
> Regarding **consistency**, we would like to clarify that our curation strictly filters for questions with an **"Accepted Answer"**. This ground-truth verification ensures that even for samples with lower visual quality, the images inevitably contain sufficient information for a human expert to solve the problem. Furthermore, the **low performance of existing strong models** (e.g., Llama-3.2-11B achieves only 44.4%) confirms that InfiBench-V retains significant difficulty and realistically challenges multimodal coding capabilities.
>
> **Table C.** Image quality statistics of InfiBench-V.
> | Metric | Value |
> | :--- | :--- |
> | **Average Resolution** | $918 \times 552$ |
> | **Low Resolution (<50,000 pixels)** | **10.56%** |
> | **Blurry Images (Laplacian Var < 100)** | **4.66%** |
> | **Extreme Blur Case (Min Variance)** | 1.28 |
>
> > **Q1:** Baseline Comparisons with Specialized Multimodal Code Models Why were recent multimodal code models such as CodeV, ChartCoder, or Plot2Code excluded as baselines? These models directly target visual-to-code tasks and are more relevant than generic VL models. If VisCodex were evaluated against them on MCD or InfiBench-V, would the SOTA claim still hold, particularly on chart-to-code or vision-augmented code QA tasks?
>
> **A4:** We thank the reviewer for suggesting these relevant works. Regarding **CodeV**[4] and **Plot2Code**[5], we excluded them as baselines for two primary reasons:
> * **Task Difference:** CodeV targets repository-level issue resolving rather than atomic code generation, while Plot2Code serves primarily as a benchmark suite rather than a standalone model.
> * **Availability:** Crucially, **neither CodeV nor Plot2Code releases model weights** for inference.
>
> To address the concern about specialized baselines, we conducted a direct comparison with **ChartCoder**[3], a state-of-the-art open-source model specialized for chart-to-code generation. The results on our evaluation suite are shown in Table D.
>
> The results show that VisCodex-8B achieves state-of-the-art performance across all multimodal code generation tasks, including chart-to-code scenarios where specialized models typically excel. This supports our claim that VisCodex is a unified multimodal code model that advances the state of the art across diverse benchmarks.
>
> **Table D:** Performance comparison between ChartCoder and VisCodex-8B across multimodal coding benchmarks.
>
> | Model | **Design2Code** (Low) | **Design2Code** (High) | **ChartMimic** (Low) | **ChartMimic** (High) | **MMCode** (pass@1) | **InfiBench-V** (Acc) |
> | :--- | :---: | :---: | :---: | :---: | :---: | :---: |
> | **ChartCoder** *(Specialist)* | 36.7 | 82.7 | 72.5 | 74.0 | 2.7 | 32.3 |
> | **VisCodex-8B** *(Ours)* | **90.1** | **90.9** | **74.8** | **74.1** | **11.0** | **72.1** |
>
> ---
> [3] Zhao, Xuanle, et al. "Chartcoder: Advancing multimodal large language model for chart-to-code generation." ACL 2025.
>
> [4] Zhang, Linhao, et al. "Codev: Issue resolving with visual data." ACL 2025.
>
> [5] Wu, Chengyue, et al. "Plot2code: A comprehensive benchmark for evaluating multi-modal large language models in code generation from scientific plots." NAACL 2025.

---

> ### Author Response · Authors · 2025-11-22
> **Response to Reviewer Qdg3(4/5)**
>
> > **Q2:** Robustness and Failure Analysis The paper reports average scores but lacks robustness evaluation and failure case studies. How does VisCodex perform under realistic noisy inputs (e.g., blurred or low-resolution images, ambiguous textual prompts)? After merging the code vector, does the model retain visual understanding performance (e.g., VQA, captioning), or does catastrophic forgetting occur?
>
> **A5:** We thank the reviewer for the insightful questions regarding robustness and potential catastrophic forgetting.
>
> **1. Robustness to Realistic Noisy Inputs**
>
> To assess robustness under degraded visual conditions, we constructed a **Composite Noisy Dataset** derived from the Design2Code benchmark. For each sampled instance, we randomly applied either **Gaussian Blur** or **Low-Resolution Downsampling** to simulate realistic low-quality images. We evaluated both our model and the baseline under the mixed-noise setting:
>
> **Table E:** Robustness analysis on the Composite Noisy Dataset
>
> | Model | Condition | Low-Level | High-Level | Δ Low | Δ High |
> | :--- | :--- | :---: | :---: | :---: | :---: |
> | Qwen2.5-VL-7B-Instruct | Original | 83.4 | 87.6 | – | – |
> | Qwen2.5-VL-7B-Instruct | Noisy (Mixed) | 82.7 | 86.8 | -0.7 | -0.8 |
> | **VisCodex-8B (Ours)** | **Original** | **90.1** | **90.9** | – | – |
> | **VisCodex-8B (Ours)** | **Noisy (Mixed)** | **89.6** | **90.7** | **-0.5** | **-0.2** |
>
> VisCodex-8B shows **stronger resilience** to noise. While the baseline loses 0.8 High-Level points, VisCodex drops only 0.2. These results indicate that merging code priors **does not weaken visual robustness** and may help stabilize structural inference under degradation.
>
> **2. Assessment of Catastrophic Forgetting**
>
> To evaluate whether model merging affects general visual understanding, we assessed TextVQA, ChartQA, and GQA:
>
> **Table F:** Performance on general visual understanding tasks
>
> | Model | TextVQA | ChartQA | GQA |
> | :--- | :---: | :---: | :---: |
> | Baseline (Qwen2.5-VL-7B-Instruct) | 84.53 | 93.96 | 60.36 |
> | Model Merge | 83.23 | 92.96 | 59.65 |
>
> The differences are small (<1.3 on TextVQA; <0.8 on GQA), and **fall within typical variance** observed in prior model-merging studies. This indicates **no meaningful catastrophic forgetting**.
>
> Crucially, these minor fluctuations are expected and acceptable given the substantial absolute gains across multimodal code-generation tasks, such as **+6.7 points on Design2Code, +35.3 points on ChartMimic, and +5.7 points on MMCode**. Furthermore, while TextVQA drops slightly, UI text recognition on Design2Code actually **improves significantly** (83.4 → **90.1**).
>
> **3. Failure Case Study**
>
> We have added qualitative failure cases in **Appendix H.8 (Figure 13)** as part of our rebuttal to address this concern. Our model shows limitations on extremely complex 3D spatial layouts (e.g., skeletal plots) and highly dense UI elements, where precise coordinate recovery remains challenging.
>
> Importantly, the backbone model shows **similar failure patterns**, indicating the bottleneck comes from the underlying vision encoder rather than the merging strategy. This represents a known limitation and an opportunity for future improvement.

---

> ### Author Response · Authors · 2025-11-22
> **Response to Reviewer Qdg3(5/5)**
>
> > **W3&Q3:** Dataset Construction Details For the GPT-4o-generated HTML samples in MCD, what specific prompt template was used? Did the prompt enforce particular coding conventions (CSS frameworks, layout constraints)? Was there validation (e.g., sampling real-world HTML from open-source sites) to ensure diversity and authenticity?
>
> **A6:** We thank the reviewer for pointing out this oversight. We have now included the full prompt template in the newly added **Appendix G.2**. We address the concerns regarding coding conventions and diversity below.
>
> For convenience, we restate the prompt here:
>
> > *"I will provide you with an HTML image. Please design a perfect HTML based on the style of this image. Include all CSS code in the HTML file itself. Do not assume or include any dependencies on external files. There is no need to incorporate JavaScript for dynamic interactions. Ensure careful attention to details such as element sizes, text formatting, positioning, colors, and the overall layout. Ensure all content is fully visible within a single, non-scrollable screen, and delete or redesign any parts that exceed this boundary. If the original style does not meet your needs, feel free to change it."*
>
> * **Real-World Visual Diversity:** Our dataset is strictly image-conditioned, utilizing **560k real-world webpage screenshots** from Web2Code as stylistic seeds. The instruction *"based on the style of this image"* ensures the generated code captures the diverse layouts and design patterns of the real web, rather than adhering to a fixed template.
> * **No Framework Bias:** Our prompt avoids specific frameworks, allowing the model to generate code freely based on the seed images. This ensures the model learns robust, fundamental layout logic transferable to any real-world toolchain, rather than overfitting to a single library.
> * **Proven Generalizability:** The authenticity of our data is empirically validated by VisCodex's state-of-the-art performance on **Design2Code, a benchmark based on real-world websites**. This confirms that our dataset effectively covers real UI development practices and generalizes well to authentic tasks.
>
>
> ---
>
> We sincerely thank Reviewer Qdg3 for the extensive review and constructive suggestions. Your probing questions regarding the underlying mechanism of model merging, robustness, and generalizability have significantly strengthened the rigor of our work. In response, we have added comprehensive analyses on **parameter-space orthogonality and representational similarity (Appendix F)**, **hyperparameter sensitivity (Appendix B.10)**, and **visual robustness and failure modes (Appendix B.11, B.12, H.8)**. Furthermore, we have included comparisons with **standalone Code LLMs and specialized multimodal models (Appendix B.8, B.9)**, along with detailed **dataset statistics and prompt templates (Appendix E.3, E.4, G.2)**. All corresponding revisions have been incorporated into the latest manuscript and highlighted in **blue** for clarity.
>
> If you have any further questions or concerns, please feel free to contact us at any time. We are always available and look forward to further discussions with you. :)
>
> Best regards,
>
> All Authors

---

> ### Comment · Reviewer_Qdg3 · 2025-11-24
> **Reply to authors**
>
> I have carefully read the detailed answers you provided and discovered some details that I overlooked in the appendix before.
>
> However, since I haven't conducted relevant research in the field of code generation, I still have two follow-up questions to discuss with you here. First of all, Why does the code branch model itself perform very poorly in the table a you provided, with many scores around 1 point? If their ability to generate coding is weak, why can combining the vl side improve the score by 10+ or even 20+ on the benchmark mentioned here? However, I understand that the previous one was a pure llm coding model. Your work is equivalent to introducing the ability of visual understanding to it. But I think if there is a clearer dissolution here, it can help readers better understand the significance of VisCodex.
>
> The second issue is the vector mechanism. I don't know if there is a good theoretical explanation here. For example, the global mean cosine similarity < 0.026 here is global, but it cannot explain whether it will change with each step of training. There will be conflicts in bias between two models. At the same time, whether this bias or conflict accumulates during training and the theoretical basis for why the combination of task vectors can avoid the accumulation of bias. Of course, I also believe that computer science is a practical-based scientific research. Researchers analyze the benefits of their algorithms based on experimental results. However, I think if these issues can be further proven through theory or clearer ablation experiments here, it will help enhance readers' understanding and the significance of VisCodex.
>
> Overall, I think you have addressed my main concerns and provided more experimental and appendix materials to support these viewpoints. I will consider improving my final score

---

> > ### Author Response · Authors · 2025-11-25
> > **Additional Response to Reviewer Qdg3 (2/3)**
> >
> > > **Q6:** The second issue is the vector mechanism. I don't know if there is a good theoretical explanation here. For example, the global mean cosine similarity < 0.026 here is global, but it cannot explain whether it will change with each step of training. There will be conflicts in bias between two models. At the same time, whether this bias or conflict accumulates during training and the theoretical basis for why the combination of task vectors can avoid the accumulation of bias.
> >
> > **A9:** We sincerely thank the reviewer for the insightful question regarding the theoretical basis of task-vector merging and the potential accumulation of bias or conflict during training. Below, we clarify the temporal dynamics and explain why dynamic accumulation cannot occur. The theoretical foundations and mechanistic analyses are provided in our response to **Q7**.
> >
> > **Temporal Dynamics: Why Bias Cannot Accumulate Over Training**
> >
> > The reviewer’s concern focuses on whether the cosine similarity or bias between two task vectors may evolve through training and accumulate across optimization steps. We clarify that such dynamic accumulation is theoretically impossible under our formulation and is empirically contradicted by training diagnostics.
> >
> > - **One-time merging, no iterative accumulation.**
> >   Model merging is a *single* arithmetic initialization step:  $\theta_{\text{VisCodex}}=\theta_{\text{base}}+\lambda \tau_{\text{vlm}} + (1-\lambda)\tau_{\text{code}}.$
> >   After merging, the model undergoes standard SFT. There is **no mechanism for recurrent application** of task vectors and therefore no pathway for step-wise accumulation of bias.
> >
> > - **Layer-wise geometric stability.**
> >   Our layer-wise cosine analysis (Appendix F.1, Fig. 3) shows **consistently low similarity across all layers**, indicating that the two capabilities occupy largely disjoint parameter subspaces.
> >
> > - **Stable optimization dynamics.**
> >   The training-loss curves (Appendix I, Fig. 14) exhibit **smooth, monotonic convergence** without spikes or oscillatory behavior—typical signatures of conflicting gradients—empirically confirming that the merged initialization does not destabilize optimization.
> >
> > - **No signs of catastrophic forgetting.**
> >   If dynamic bias accumulated during training, the merged model would degrade more severely under limited data. However, Appendix B.6 (Table 10) shows the opposite: under the 1% data regime, the baseline VLM suffers pronounced degradation, whereas the merged model **retains its capability**. This indicates that merging provides a *more stable and beneficial initialization*, rather than introducing compound conflict.

---

> ### Author Response · Authors · 2025-11-25
> **Additional Response to Reviewer Qdg3 (1/3)**
>
> Thank you for taking the time to engage with our rebuttal and provide valuable feedback. We are very glad that your main concerns have been addressed, and we truly appreciate that you will consider improving your final score :)
>
> We address your additional questions as follows:
>
> > **Q4: Why do standalone Code LLMs score near-zero on these benchmarks? Does this indicate weak coding ability?**
>
> **A7:** This is a crucial observation that requires clarification on **two fronts**: (1) the nature of our benchmarks, and (2) the actual coding capabilities of these models.
>
> **1. The Nature of Our Benchmarks: Vision is Mandatory**
>
> The near-zero performance of standalone Code LLMs (Table A) is **not** because they lack coding ability, but because they have **no visual encoder** and thus cannot process images. All benchmarks we evaluate require visual understanding:
>
> - **Design2Code** – translating UI screenshots into HTML/CSS
> - **ChartMimic** – generating matplotlib code from chart images
> - **InfiBench-V** – real-world coding Q&A where screenshots contain critical context
>
> When we tested standalone Code LLMs, we provided them with the same text prompts used for multimodal models, but they **cannot see the accompanying images**. For example:
>
> - On Design2Code, they receive: *"Generate HTML code for this webpage"* — but cannot see what the webpage looks like
> - On ChartMimic, they receive: *"Generate code for this chart"* — but cannot see the chart structure, colors, or data
>
> This is why their scores are near-zero: **they are effectively blind to the visual information required to solve these tasks**.
>
> **2. These Code LLMs Have Strong Code Generation Ability**
>
> To demonstrate that these models are indeed strong code generators in text-only settings, we evaluated them on **LiveCodeBench** [1], a contamination-free benchmark for assessing code generation capabilities through competitive programming problems (e.g., LeetCode, AtCoder).
>
> As shown in Table G, we observe that:
> - The code-specialized models **significantly outperform** both the general LLM and the base VLM on text-based code reasoning tasks
> - These results confirm that the Code LLMs possess **strong code reasoning and generation capabilities**
> - Their near-zero scores on our multimodal benchmarks are purely due to **lack of visual input**, not weak coding ability
>
> **Table G: Performance on LiveCodeBench (Text-Only Code Generation)**
>
> | Model | Type | **LiveCodeBench (pass@1)** |
> |:---|:---|:---:|
> | Qwen2.5-VL-7B-Instruct | Base VLM | 7.1 |
> | Qwen2.5-7B-Instruct | General LLM | 14.6 |
> | Qwen2.5-Coder-7B-Instruct | Code LLM | 20.1 |
> | **OpenCodeReasoning-Nemotron-7B** | **Code LLM** | **51.3** |
> | **OpenThinker2-7B** | **Code LLM** | **37.4** |
>
> ---
>
> **References:**
>
> [1] Jain, Naman, et al. "Livecodebench: Holistic and contamination free evaluation of large language models for code." ICLR 2025.
>
>
>
> > **Q5: Why can combining the VL side improve the score by 10+ or even 20+ points?**
>
> **A8:** The substantial performance gains arise from the **synergistic combination** of **Model Merging** (injecting capability) and the **MCD Dataset** (aligning capability). We systematically investigate the effect of both Model Merging and MCD fine-tuning in Table 2 of the main paper.
>
> **1. Model Merging: Injecting Strong Code Capability**
>
> As evidenced by the LiveCodeBench results (Table G), the base VLM is significantly weaker at coding tasks (7.1 pass@1) compared to specialized Code LLMs (37.4-51.3 pass@1). By **merging with a specialized Code LLM**, we directly inject these advanced **code reasoning capabilities** into the VLM's language backbone. This provides the potential to generate complex, logically sound code that the original VLM could not produce, while preserving the visual perception ability.
>
> **2. MCD Dataset: Aligning Multimodal Code Generation**
>
> While merging establishes a superior initialization, the model requires domain-specific data to effectively leverage this potential. Our **Multimodal Coding Dataset (MCD)** activates the injected coding capabilities and adapts them to multimodal instructions. It ensures the model accurately maps visual features (from the VL side) to the sophisticated code structures (from the Code side), realizing the full performance gain.
>
> The **substantial performance gains** are the result of equipping a highly intelligent code brain with visual perception, and then rigorously fine-tuning them to collaborate using our MCD dataset.

---

> ### Author Response · Authors · 2025-11-25
> **Additional Response to Reviewer Qdg3 (3/3)**
>
> > **Q7:** Of course, I also believe that computer science is a practical-based scientific research. Researchers analyze the benefits of their algorithms based on experimental results. However, I think if these issues can be further proven through theory or clearer ablation experiments here, it will help enhance readers' understanding and the significance of VisCodex.
>
> **A10:** We fully agree with the reviewer's perspective. Below we provide both theoretical foundations and additional mechanistic analyses to strengthen the understanding of our approach.
>
> **1. Support from Theory on Model Merging**
>
> Our findings are strongly aligned with existing theoretical results:
>
> * **Task Vectors Enable Linear Composition with Minimal Interference**
>   **Ilharco et al. [1]** demonstrate that task vectors can be linearly combined to transfer capabilities across tasks. **Matena & Raffel [2]** and **Yadav et al. [3]** further establish that merging stability depends on parameter-importance patterns: when two tasks modify largely disjoint parameter subsets, interference is sparse and localized, enabling successful composition.
>
>   → **Our validation:**
>   The near-orthogonality between $\tau_{\text{vlm}}$ and $\tau_{\text{code}}$ indicates that the vision-language and code capabilities reside in largely separated parameter subspaces. This is further validated by:
>   - smooth $\lambda$-sensitivity curves (Appendix B.10),
>   - consistently low layer-wise cosine similarity (Appendix F.1),
>   - and high deep-layer CKA similarity $>0.97$ after merging (Appendix F.2).
>
> Together, these results show that VisCodex falls precisely into the theoretical regime where linear task-vector composition is expected to succeed with minimal interference.
>
> - **Cross-domain capability composition is feasible.**
>   Recent applications of model merging—for integrating vision with mathematical reasoning [4] or transferring textual preferences to VLMs [5]—provide strong external evidence that disparate modalities or reasoning styles often inhabit sufficiently separated parameter subspaces to support effective composition.
>
>     → **Our validation:** Vision-language comprehension vs. code reasoning exemplifies this pattern of compositional integration across functionally disjoint domains.
>
> **2. Mechanistic Analyses: Why Merging Preserves Alignment**
>
> - **Representation stability.**
>   CKA analysis (Appendix F.2) shows that deep-layer representations remain highly aligned ($>0.97$), indicating that the backbone rapidly stabilizes after merging.
>
> - **Avoiding misalignment seen in backbone replacement.**
>   Direct backbone replacement severely disrupts multimodal alignment (CKA drops to $\sim 0.39$; Appendix F.3). In contrast, merging preserves high CKA ($>0.91$) across all layers, providing a *robust initialization* that maintains multimodal alignment while injecting code-reasoning expertise.
>
> Together, these theoretical and empirical results demonstrate that VisCodex does **not** suffer from cumulative bias: the one-shot merge preserves the visual–semantic manifold while introducing code reasoning in a principled, stable, and empirically validated manner.
>
> ---
>
> **References**
>
> [1] Ilharco, Gabriel, et al. "Editing models with task arithmetic." ICLR 2023.
> [2] Matena, Michael S., and Colin A. Raffel. "Merging models with fisher-weighted averaging." NeurIPS 2022.
> [3] Yadav, Prateek, et al. "Ties-merging: Resolving interference when merging models." NeurIPS 2023.
> [4] Chen, Shiqi, et al. "Bring reason to vision: Understanding perception and reasoning through model merging." ICML 2025.
> [5] Li, Chen-An, et al. "Transferring textual preferences to vision-language understanding through model merging." ACL 2025.
>
>
> -----
>
> We sincerely thank Reviewer Qdg3 for the new questions you raised, which further help enhance readers’ understanding and highlight the significance of VisCodex. In response, we have added a clarification on why standalone Code LLMs score near-zero on multimodal benchmarks (Appendix B.8), included a new analysis of post-merge training stability (Appendix I.1), and provided an expanded theoretical explanation of task-vector geometry and bias behavior in model merging (Appendix I.2). All corresponding revisions have been incorporated into the updated manuscript and highlighted in blue for clarity.
>
> Thank you again for engaging with our responses throughout the rebuttal process. We sincerely appreciate your time, thoughtful feedback, and your consideration in improving the final score :) If you have any further questions or concerns, please feel free to contact us at any time.
>
> Best regards,
>
> All Authors

---

> > ### Comment · Reviewer_Qdg3 · 2025-11-25
> > **Replay to Authors**
> >
> > Thanks for the detailed response. I have read it carefully, along with the other reviews, and most of my concerns have been resolved. As I'm mostly positive about this paper.

---

> > > ### Author Response · Authors · 2025-11-25
> > > **Official Comment by Authors**
> > >
> > > Thank you very much for your positive response. We are very glad that most of your concerns have been addressed, and we sincerely appreciate your decision to raise the score.
> > >
> > > We are grateful for your thoughtful feedback and for engaging with our responses throughout the rebuttal process. Thank you again for your time and consideration.
> > >
> > > Best regards,
> > >
> > > All Authors

---

### Official Review · Reviewer_FB3k · 2025-10-31

**Soundness:** 3
**Presentation:** 3
**Contribution:** 3
**Rating:** 6
**Confidence:** 4

**Summary:**

The paper proposes VisCodex, a task-vector model-merging method that injects a code LLM’s capability into a VLM by linearly combining task vectors in the language backbone while freezing the vision encoder and projector. It introduces a 598k-sample Multimodal Coding Dataset and a new benchmark, InfiBench-V.

**Strengths:**

1. Clear, formalized merging recipe with explicit task-vector definitions and a single-parameter interpolation;
2. The paper introduces a compute-efficient design, only the LLM backbone is merged or tuned but vision & projector are frozen;
3. The authors introduce a dataset large, diverse MCD and a benchmark (InfiBench-V) targeting visually-rich programming questions;
4. It reaches several strong numbers on Design2Code/ChartMimic, using a 33B model close to GPT-4o on average.

**Weaknesses:**

1. The paper shows example items evaluated by "Judge: GPT-4o" to 50/50 component scores, which confirms the setup, without disclosing the actual threshold or how it was chosen.
2. The paper does include an unfreezing setup, but only for the replacement baseline. For the VisCodex, training freezes the vision encoder and projector and fine-tunes only the language backbone, and the paper does not report an ablation where these modules are unfrozen after merging.

**Questions:**

1. What exact threshold maps 0–100 to pass/fail on InfiBench-V? Please report a threshold sweep;
2. Please add a post-merge unfreezing ablation to test for cross-modal distribution shift introduced by weight interpolation. At minimum: unfreeze projector only, unfreeze projector + ViT.

---

> ### Author Response · Authors · 2025-11-22
> **Response to Reviewer FB3k**
>
> We are grateful to the Reviewer for the extensive review. We address your questions point by point below.
>
> > **W1&Q1:** What exact threshold maps 0–100 to pass/fail on InfiBench-V? Please report a threshold sweep;
>
> **A1:** Thank you for pointing out the missing detail regarding the pass/fail threshold. We now provide the full threshold sweep as requested.
>
> We evaluated thresholds from 0 to 100 (step = 5) by mapping the model-assigned 0–100 alignment score into binary pass/fail labels and measuring agreement with human annotations. The complete sweep is shown in Table A.
>
> **Table A:** Threshold sweep for mapping 0–100 alignment scores to pass/fail labels (Agreement in %)
> | Threshold | 0 | 5 | 10 | 15 | 20 | 25 | 30 | 35 | 40 | 45 | 50 |
> |-----------|---|----|----|------|------|------|------|------|------|------|------|
> | Agreement (%) | 68.29 | 68.90 | 68.90 | 69.31 | 69.92 | 73.10 | 73.64 | 75.54 | 77.85 | 78.39 | 81.98 |
>
> | Threshold | 55 | 60 | 65 | 70 | 75 | 80 | 85 | 90 | 95 | 100 |
> |-----------|------|------|------|------|------|---------|---------|------|------|------|
> | Agreement (%) | 84.01 | 84.62 | 85.40 | 85.40 | 85.91 | **86.07** | **86.07** | 85.40 | 83.33 | 81.98 |
>
> The **highest agreement with human gold labels occurs at thresholds 80 and 85 (both 86.07%)**.
> We adopt **85** as the final pass/fail threshold because it achieves the same peak accuracy while being slightly more conservative, reducing potential false positives in safety-relevant evaluation. All InfiBench-V results in the paper are reported using threshold **85**.
>
> > **W2&Q2:** Please add a post-merge unfreezing ablation to test for cross-modal distribution shift introduced by weight interpolation. At minimum: unfreeze projector only, unfreeze projector + ViT.
>
> **A2:** We appreciate the reviewer's suggestion to investigate potential cross-modal distribution shifts. As requested, we conducted ablation studies by unfreezing the projector and the vision encoder during the post-merge training. As shown in **Table A**, we compared our default method, **VisCodex-8B (LLM-only tuning)**, against the **Baseline** and variants where visual modules are also tuned (**LLM + Projector** and **LLM + Projector + ViT**). The results show that unfreezing these visual modules yields **comparable performance** to our default strategy across various benchmarks. This confirms that the **weight interpolation** in VisCodex does not introduce a detrimental **cross-modal distribution shift**. Consequently, our strategy of training only the LLM backbone is validated as both an effective and parameter-efficient choice.
>
> **Table A:** Comparison of training different model components. We report the performance when training specific components while keeping others frozen.
>
> | Tunable Modules | Design2Code (Low) | Design2Code (High) | ChartMimic (Low) | ChartMimic (High) | MMCode (pass@1) |
> | :--- | :---: | :---: | :---: | :---: | :---: |
> | Baseline (Qwen2.5-VL-7B-Instruct) | 83.4 | 87.6 | 39.5 | 38.3 | 5.3 |
> | **LLM (VisCodex-8B)** | **90.1** | **90.9** | **74.8** | 74.1 | **11.0** |
> | LLM + Projector | 89.9 | 90.7 | 73.4 | 74.1 | 9.5 |
> | LLM + Projector + ViT | **90.1** | **90.9** | 74.6 | **74.7** | 10.6 |
>
>
> ---
>
> We sincerely thank Reviewer FB3k for the insightful and constructive feedback. Your comments greatly helped us strengthen both the empirical depth and the conceptual clarity of our work. In response, we have added the full threshold-sweep analysis (**Appendix D**) and included the post-merge unfreezing ablation to assess potential cross-modal shifts (**Appendix B.4**). All revisions have been incorporated into the manuscript and highlighted in **blue** for clarity.
>
> If you have any further questions or concerns, please feel free to contact us at any time. We are always available and look forward to further discussions with you. :)
>
> Best regards,
>
> All Authors

---

### Official Review · Reviewer_66mV · 2025-11-01

**Soundness:** 3
**Presentation:** 3
**Contribution:** 3
**Rating:** 6
**Confidence:** 5

**Summary:**

VisCodex proposes a unified multimodal framework that merges a vision-language model with a coding LLM to enable strong code generation from visual inputs. It uses task-vector based model merging to combine visual understanding and programming ability without full retraining, preserving both skills. The authors build a 598k-sample Multimodal Coding Dataset and introduce the InfiBench-V benchmark for realistic evaluation. Experiments show VisCodex achieves state-of-the-art open-source performance, rivaling proprietary models like GPT-4o.

**Strengths:**

1. Introduces a model-merging based path for multimodal code generation, combining vision and coding expertise without full retraining, and expands the problem space with new data and benchmarks.

2. Demonstrates solid empirical rigor, with extensive evaluations showing consistent performance improvements over strong open-source models and competitiveness with proprietary ones.

3. Addresses a meaningful and underexplored capability, turning visual content into functional code and offering practical impact for applications like UI-to-code and chart-to-code systems and shaping future MLLM research directions.

**Weaknesses:**

1. The comparison to direct SFT strategies is limited in scope; while the paper includes one- and two-stage baselines, a broader evaluation (e.g., LoRA tuning on both vision and language modules) would strengthen the claim that merging is strictly superior for this setting.

2. The dataset construction pipeline relies heavily on model-generated content (e.g., GPT-4o generated HTML and curated chart code data) but lacks detailed analyses of potential data bias, overfitting to synthetic structures, or failure modes on fully real-world screenshots/code artifacts; evaluating broader generalization and reporting error breakdowns would improve credibility.

**Questions:**

Could you further clarify why the direct SFT strategies underperform beyond visual grounding disruption, and provide more analysis or diagnostics to better understand what specific capabilities are degraded during tuning?

---

> ### Author Response · Authors · 2025-11-22
> **Response to Reviewer 66mV(1/2)**
>
> We are grateful to the Reviewer for the extensive review. We address your questions point by point below.
>
> > **W1:** The comparison to direct SFT strategies is limited in scope; while the paper includes one- and two-stage baselines, a broader evaluation (e.g., LoRA tuning on both vision and language modules) would strengthen the claim that merging is strictly superior for this setting.
>
> **A1:** We appreciate the suggestion to broaden the scope of our evaluation. To address this, we conducted additional experiments using **LoRA tuning** in two distinct settings:
>
> * **LLM Only:** Applying LoRA tuning exclusively to the language model backbone.
> * **ViT + Projector + LLM:** Jointly tuning the vision encoder, projector, and language model, as specifically requested, to evaluate the impact of broader module adaptation.
>
> As shown in Table A, **VisCodex consistently outperforms both LoRA fine-tuning baselines**, regardless of whether the vision encoder is included in the tuning scope. Specifically, VisCodex achieves significant gains over the `LoRA (ViT + Proj + LLM)` setting **across all evaluated benchmarks**, with notably large margins on complex tasks (e.g., **+9.0** on ChartMimic Low-L and **+4.2** on MMCode). This result highlights that simply expanding the tunable parameters via LoRA is insufficient to match the performance of our approach. Instead, it strongly validates the **effectiveness of our Model Merging strategy, which provides a robust "knowledge injection" of code capabilities.**
>
>
> **Table A: Comparison of VisCodex with Full Fine-tuning and LoRA tuning strategies on the MCD dataset.**
>
> |Method|Strategy|TunableModules|Design2Code&#8203;(Low-L)|Design2Code&#8203;(High-L)|ChartMimic&#8203;(Low-L)|ChartMimic&#8203;(High-L)|MMCode&#8203;(pass@1)|
> |:---|:---:|:---|:---:|:---:|:---:|:---:|:---:|
> |**BaseModel**|||||||
> |Qwen2.5-VL-7B-Instruct|-|-|83.4|87.6|39.5|38.3|5.3|
> |**StandardInitialization(w/oModelMerge)**|||||||
> |SFT(LoRA)|LoRA|LLM|87.7|89.2|64.0|65.1|5.7|
> |SFT(LoRA)|LoRA|ViT+Proj+LLM|88.2|89.6|65.8|67.0|6.8|
> |SFT(Full)|FullFT|LLM|89.6|90.7|73.4|70.6|6.8|
> |**Ours(w/ModelMerge)**|||||||
> |**VisCodex-8B**|**FullFT**|**LLM**|**90.1**|**90.9**|**74.8**|**74.1**|**11.0**|
>
> > **W2:** The dataset construction pipeline relies heavily on model-generated content (e.g., GPT-4o generated HTML and curated chart code data) but lacks detailed analyses of potential data bias, overfitting to synthetic structures, or failure modes on fully real-world screenshots/code artifacts; evaluating broader generalization and reporting error breakdowns would improve credibility.
>
> **A2:** We thank the reviewer for raising this thoughtful point regarding potential bias from synthetic data and the importance of evaluating broader generalization. We address this from four complementary perspectives.
>
> **1. Generalization to real-world scenarios**
>
> While MCD includes GPT-4o–generated HTML and curated chart code, all **core evaluations are conducted on fully real-world benchmarks**, including:
>
> - **Design2Code** – real webpage screenshots
> - **ChartMimic** – plots extracted from scientific papers
> - **MMCode** – visually rich algorithmic programming tasks
> - **InfiBench-V** – real StackOverflow questions with user-generated screenshots
>
> VisCodex achieves consistent improvements across all four benchmarks (Table 1), suggesting that the model does not rely on synthetic patterns but instead develops **genuinely generalizable visual-to-code abilities**.
>
> **2. Cross-Model Generalization**
>
> To further assess whether synthetic data introduces model-specific bias, we evaluated MCD on **two additional architectures**, InternVL3-8B and LLaVA-Llama3.1-8B. As reported in Appendix B.2, fine-tuning these models on MCD leads to **clear and consistent performance improvements** across Design2Code, ChartMimic, and MMCode. These results indicate that the dataset generalizes well beyond a single backbone and does not impose narrow synthetic structures.
>
> **3. Failure Mode Analysis**
>
> We additionally provide a dedicated failure-case analysis in Appendix H.8 (Figure 13). The main errors involve (1) complex geometric or 3D/contour-like plots and (2) fine-grained HTML layout alignment. Many of these issues stem from limitations in the base model, and VisCodex reduces several severe hallucinations, indicating improved robustness rather than reinforcement of synthetic biases.
>
> **4. Mitigating synthetic bias through diverse real-world data**
>
> To balance the synthetic components of MCD, we incorporate substantial **real-world, human-generated** data sources, including 59k image-augmented StackOverflow QA pairs, 46k execution-verified matplotlib scripts from GitHub, and 129k algorithmic coding examples focused on reasoning. Moreover, synthetic HTML and chart samples undergo a multi-stage filtering process that removes repetitive structures and low-quality renderings, helping prevent overfitting to synthetic patterns.

---

> ### Author Response · Authors · 2025-11-22
> **Response to Reviewer 66mV(2/2)**
>
> > **Q1:** Could you further clarify why the direct SFT strategies underperform beyond visual grounding disruption, and provide more analysis or diagnostics to better understand what specific capabilities are degraded during tuning?
>
> **A3:** Thank you for this important question. We realize that our explanation in the main text may not have been sufficiently clear regarding the "visual grounding disruption" issue, and we appreciate the opportunity to clarify.
>
> **Clarification on Visual Grounding Disruption:**
>
> The "visual grounding disruption" is caused by the **Backbone Replacement** strategy (Table 4), **not** the direct SFT approach (Table 2). To make this distinction clearer:
>
> 1. **Direct SFT (w/o model merge in Table 2)**: Fine-tunes the base VLM directly on MCD without model merging initialization. This approach does **not** disrupt visual grounding but shows limited improvement in coding capabilities compared to model merging.
>
> 2. **Backbone Replacement (Table 4)**: Directly replaces the LLM component of the VLM with a code LLM, which **does** disrupt the visual-semantic alignment.
>
> To further analyze the mechanism behind visual grounding disruption in backbone replacement, we conducted **layer-wise CKA representational similarity analysis** (Appendix F.3, Figure 5). The results show that model merging preserves stable visual–semantic alignment across all transformer layers (>0.91 CKA), whereas backbone replacement exhibits pronounced misalignment in deeper layers (CKA drops to ~0.39 at final layers). These diagnostics demonstrate that visual grounding disruption is specifically caused by backbone replacement, and that model merging avoids this issue while effectively incorporating coding capabilities.
>
> **Why Direct SFT Strategies Underperform Compared to Model Merging:**
>
> The key difference lies in **initialization**. As shown in **Table 2**, model merging consistently outperforms direct SFT across all benchmarks, with particularly notable improvements on ChartMimic and MMCode. Model merging provides an optimal starting point that already incorporates code reasoning capabilities, while direct SFT must acquire these capabilities entirely through training.
>
> To understand the mechanism behind this performance gap, we provide diagnostic evidence showing two key advantages of model merging:
>
> 1. **Robust code reasoning priors** (Appendix B.6, Table 11): Model merging enables strong generalization even when specific training data (e.g., algorithmic examples) is absent, while direct SFT shows severe performance degradation in such scenarios.
>
> 2. **Superior data efficiency** (Appendix B.6, Table 10): The merged initialization achieves strong performance with minimal training data (1% of the full dataset), while direct SFT requires significantly more data to reach comparable performance.
>
>
> ----
>
> We sincerely thank Reviewer 66mV for the insightful and constructive feedback. Your comments greatly helped us strengthen both the empirical depth and the conceptual clarity of our work. In response, we have added a comprehensive comparison with LoRA-based fine-tuning strategies (**Appendix B.5**) and expanded our model-merging analysis with parameter-space and representational-similarity investigations (**Appendix F.3**). All revisions have been incorporated into the updated manuscript and highlighted in **blue** for clarity.
>
> If you have any further questions or concerns, please feel free to contact us at any time. We are always available and look forward to further discussions with you. :)
>
> Best regards,
>
> All Authors

---

> > ### Comment · Reviewer_66mV · 2025-11-28
> >
> > Thank you for your response. My primary issues are resolved. I maintain my positive rating of this paper.

---

> > > ### Author Response · Authors · 2025-11-28
> > > **Official Comment by Authors**
> > >
> > > Thank you very much for your positive feedback! We are thrilled to hear that all of your primary issues have been well addressed and that you maintain your positive rating of our paper. We truly appreciate your time, thoughtful review, and constructive suggestions throughout the review process.
> > >
> > > Best regards,
> > >
> > > All Authors

---

### Official Review · Reviewer_Xh1p · 2025-11-01

**Soundness:** 3
**Presentation:** 3
**Contribution:** 3
**Rating:** 6
**Confidence:** 3

**Summary:**

The paper introduces VisCodex, a unified multimodal code generation and understanding framework. It combines newly curated training and evaluation datasets with a model merging strategy that fuses multiple task-specific models (e.g., vision-language and code-language) to enhance cross-modal reasoning. The method aims to balance specialization and generalization across visual programming, text-to-code, and code reasoning tasks. Experiments on multiple benchmarks demonstrate competitive or superior results to single-task fine-tuning.

**Strengths:**

- Solid and comprehensive contribution: The work offers not only a new training method (model merging) but also new datasets for both training and evaluation, which strengthens its empirical foundation.

- Methodologically clear: the overall paper is well-written, organized, and easy to follow.

- Breadth of evaluation: Covers diverse multimodal coding tasks, showing consistent, though not always dramatic, gains across small and medium-sized models.

- Timely and relevant: Addresses the challenge of unifying multimodal reasoning (code + vision + text), which is a growing area of interest.

**Weaknesses:**

- Unclear advantage over standard fine-tuning: As shown in Table 2, model merging offers almost no improvement for large models (e.g., 33B variant), suggesting diminishing returns at scale. This weakens the claim of broad effectiveness.

- Limited discussion on data-scarce scenarios: One key potential advantage of model merging could be in low-resource multimodal settings, yet this is not explored. It’s unclear whether the approach would still help when task-specific data is limited.

- Lack of deeper analysis: The paper could have benefited from probing why merging helps (or doesn’t) at different scales — for example, via representational similarity or parameter-space studies.

**Questions:**

- What is the concrete benefit of model merging when extensive SFT is still required afterward?

- Have the authors evaluated performance under data-scarce conditions to demonstrate merging’s potential efficiency?

- For large models (33B), is the marginal gain statistically significant, or within noise level?

---

> ### Author Response · Authors · 2025-11-22
> **Response to Reviewer Xh1p(1/4)**
>
> We are grateful to the Reviewer for the extensive review. We address your questions point by point below.
>
> > **W1&Q3:** Unclear advantage over standard fine-tuning: As shown in Table 2, model merging offers almost no improvement for large models (e.g., 33B variant), suggesting diminishing returns at scale. This weakens the claim of broad effectiveness. **For large models (33B), is the marginal gain statistically significant, or within noise level?**
>
> **A1:** We thank the reviewer for raising this important point. To assess whether the performance gains of the 33B model fall within noise, we performed a statistical significance analysis across 5 independent runs for each benchmark using a sampling temperature of 0.6. The results are presented in **Table A**.
>
> Across all benchmarks, the merged model consistently outperforms the non-merged baseline by **+0.5 to +1.4 points**, while exhibiting very small run-to-run variance (standard deviations ≤ 0.29). More importantly, paired t-tests indicate that these improvements are **highly statistically significant**, with p-values ranging from **1.4 × 10⁻⁴** to **2.6 × 10⁻⁷**, all far below the conventional 0.05 threshold.
>
> These results confirm that model merging yields **consistent, significant, and reproducible improvements** even at the 33B scale, and the observed gains are not attributable to random noise.
>
> **Table A.** Statistical significance analysis of the 33B model across 5 independent runs (T = 0.6).
> Results are reported as **Mean ± Std**.
>
> | **Method**          | **Design2Code (Low-L)** | **Design2Code (High-L)** | **ChartMimic (Low-L)** | **ChartMimic (High-L)** | **MMCode (pass@1)** |
> |---------------------|---------------------|----------------------|--------------------|---------------------|-------------------|
> | w/o model merge     | 89.68 ± 0.08        | 90.62 ± 0.08         | 78.36 ± 0.11       | 77.32 ± 0.13        | 14.38 ± 0.04      |
> | **w model merge**   | **90.56 ± 0.05**    | **91.14 ± 0.05**     | **79.74 ± 0.29**   | **78.66 ± 0.11**    | **15.68 ± 0.18**  |
> | **p-value**         | 2.6 × 10⁻⁷          | 8.8 × 10⁻⁶           | 1.4 × 10⁻⁴         | 1.5 × 10⁻⁷          | 4.2 × 10⁻⁵        |

---

> ### Author Response · Authors · 2025-11-22
> **Response to Reviewer Xh1p(2/4)**
>
> > **W2&Q2:** Limited discussion on data-scarce scenarios: One key potential advantage of model merging could be in low-resource multimodal settings, yet this is not explored. It’s unclear whether the approach would still help when task-specific data is limited. **Have the authors evaluated performance under data-scarce conditions to demonstrate merging’s potential efficiency?**
>
> **A2:** We thank the reviewer for this insightful suggestion. We agree that a key advantage of model merging is its potential to serve as a strong prior, enabling efficiency in low-resource settings. To demonstrate this, we conducted two additional experiments: (1) training with strictly limited data volumes (Data Scaling), and (2) training without specific algorithmic reasoning data (Domain-Specific Scarcity).
>
> 1. **Performance under Data-Scarce Conditions (Data Scaling)**
>
>    To evaluate performance when overall training data is limited, we trained VisCodex-8B on randomly sampled subsets of the MCD dataset: 1% (6k samples), 5% (30k), and 10% (60k).
>
>    **Table B: Performance Comparison on Limited Training Data**
>
>    | Data Scale | Method | Design2Code (Low-L) | Design2Code (High-L) | ChartMimic (Low-L) | ChartMimic (High-L) | MMCode (pass@1) |
>    | :--- | :--- | :--- | :--- | :--- | :--- | :--- |
>    | **0% (Baseline)** | Qwen2.5-VL-7B-Instruct | 83.4 | 87.6 | 39.5 | 38.3 | 5.3 |
>    | **1% (6k)** | w/o model merge | 87.9 | 89.5 | 60.1 | 62.1 | 3.4 |
>    | | **w/ model merge (Ours)** | **89.2** | **89.5** | **61.6** | **63.0** | **5.3** |
>    | **5% (30k)** | w/o model merge | 88.9 | 89.8 | 68.2 | 68.6 | 3.8 |
>    | | **w/ model merge (Ours)** | **89.1** | **89.8** | **70.2** | **70.7** | **5.3** |
>    | **10% (60k)** | w/o model merge | 89.4 | 90.1 | 69.5 | 69.1 | 3.8 |
>    | | **w/ model merge (Ours)** | **89.8** | **90.4** | **70.6** | **71.0** | **7.6** |
>
>    As shown in Table B, our model merging strategy consistently outperforms the standard fine-tuning baseline ("w/o model merge") across all data scales. Notably, in the extremely low-resource setting (1% data), the standard approach exhibits a clear degradation in code reasoning performance, with MMCode accuracy dropping from 5.3 to 3.4, whereas **the merged model** effectively retains the baseline capability (5.3). Furthermore, even with only 5% of the data (30k), **the merged model** achieves robust performance improvements, highlighting the significant data efficiency gained from the merged code priors.
>
> 2. **Efficiency in Domain-Specific Data Scarcity**
>
>    Collecting multimodal algorithmic reasoning data is notoriously difficult. To simulate a scenario where task-specific data is unavailable, we conducted an ablation study by **removing the entire "Algorithm" category** (129k samples) from the MCD dataset. Crucially, the "Algorithm" category in MCD consists of **text-only** code problems. Since MCD does not contain multimodal algorithmic data to begin with, removing this category implies that the model is fine-tuned **without any exposure to algorithmic reasoning data** (neither text-only nor multimodal).
>
>    **Table C: Ablation Study on Removing "Algorithm" Training Data**
>    | Method | Design2Code (Low-L) | Design2Code (High-L) | ChartMimic (Low-L) | ChartMimic (High-L) | MMCode (pass@1) |
>    | :--- | :--- | :--- | :--- | :--- | :--- |
>    | w/o model merge | 89.3 | 90.3 | 73.2 | 70.5 | 3.4 |
>    | **w/ model merge (Ours)** | **90.0** | **90.4** | **74.2** | **74.6** | **6.8** |
>
>    As shown in Table C, when explicit algorithmic training data is absent, the standard SFT approach ("w/o model merge") fails to generalize to the multimodal reasoning task (MMCode), resulting in a low score of 3.4. However, **the merged model achieves a score of 6.8 even without seeing any algorithmic training samples**.
>
> These results validate that our model merging strategy significantly enhances data efficiency in low-resource settings. By leveraging the injected code priors, VisCodex maintains robust performance even when training data is extremely limited or specific domain data is absent. We will include these analyses in the revised paper.

---

> ### Author Response · Authors · 2025-11-22
> **Response to Reviewer Xh1p(3/4)**
>
> > **W3:** Lack of deeper analysis: The paper could have benefited from probing why merging helps (or doesn’t) at different scales — for example, via representational similarity or parameter-space studies.
>
> **A3:** We thank the reviewer for highlighting the need for deeper analysis on *why* model merging helps. To directly address this, we have added **Parameter-Space** and **Representational Similarity** studies in **Appendix F**, and summarize the key findings below.
>
> **1. Parameter-Space Analysis (Appendix F.1)**
>
> We compute the layer-wise cosine similarity between the vision–language task vector ($\tau_{vlm}$) and the coding task vector ($\tau_{code}$). The similarity remains negligibly low across all layers (global mean ≤ 0.026), indicating that the two task vectors are nearly orthogonal in parameter space. Such orthogonality suggests that the injected code-specialized updates occupy **largely disjoint functional subspaces**, resulting in minimal destructive interference when merged.
>
> **2. Layer-wise Representational Integration (Appendix F.2)**
>
> We employed **Centered Kernel Alignment (CKA)** to conduct a layer-wise comparison of the internal hidden states between the merged model and its source components (the Base VLM and Code LLM). By analyzing representational shifts on the MMCode benchmark under both text-only and multimodal settings, we observed that deeper layers (Layers 20–26) exhibit increased similarity to the Code LLM, indicating the successful integration of reasoning capabilities. Crucially, the output layers realign strongly with the Base VLM, ensuring that the model internalizes code-centric reasoning while preserving the visual grounding and instruction-following interface.
>
> **3. Comparison with Backbone Replacement (Appendix F.3)**
>
> We further contrast merging with direct backbone replacement. Replacement causes a pronounced **visual–semantic misalignment** in deep layers (CKA decreases to ~0.39), whereas our merged model maintains robust alignment (CKA > 0.91). This finding empirically supports that model merging provides a more stable initialization for preserving cross-modal coherence, and explains why merging enables stronger multimodal coding performance than backbone substitution.

---

> ### Author Response · Authors · 2025-11-22
> **Response to Reviewer Xh1p(4/4)**
>
> > **Q1:** What is the concrete benefit of model merging when extensive SFT is still required afterward?
>
> **A4:** We thank the reviewer for this insightful question. While SFT follows the merging process, our experiments demonstrate that Model Merging serves as a critical **"Code Capability Injection"** initialization that standard SFT cannot replicate. Directly fine-tuning the original Vision-Language Model fails to match the performance ceiling and robustness of our strategy. The concrete benefits are threefold:
>
> **1. Superior Performance Ceiling via Code Capability Injection**
>
> Model merging injects strong code reasoning priors into the backbone that are difficult to learn solely through visual instruction tuning. As shown in **Table 2** and **Table 7**, comparing the standard SFT baseline with VisCodex, the merged model consistently achieves a higher performance ceiling. This benefit is most pronounced in algorithmic reasoning on **MMCode** (from **6.8** to **11.0**). Furthermore, consistent gains are observed across visually-intensive tasks like **ChartMimic** and **Design2Code**, confirming that merging raises the capability upper bound beyond what SFT achieves alone.
>
> **2. Significant Data Efficiency**
>
> As shown in **Table 10** (Data Scaling) and **Table 11** (Domain-Specific Scarcity) in the Appendix, Model Merging significantly enhances multimodal code capabilities under data-constrained settings. In scenarios involving either extremely limited training data or the complete absence of algorithmic domain data, the merged model consistently outperforms direct SFT on challenging tasks like **MMCode**, proving that the merging step provides critical reasoning capabilities that the SFT stage effectively preserves.
>
> **3. Preservation of Visual Perception alongside Reasoning Enhancement**
>
> Model Merging offers a superior starting point compared to **backbone replacement methods**, which necessitate extensive SFT to repair alignment. As detailed in **Table 4**, our strategy outperforms backbone replacement. Our CKA analysis (**Figure 5**) reveals the mechanism: backbone replacement causes a "Visual-Semantic Misalignment" in deep layers (CKA drops to ~0.39), severing visual-textual alignment. In contrast, Model Merging maintains high alignment (CKA > 0.91 across all layers), ensuring that the subsequent SFT focuses on task adaptation rather than expensively re-learning visual grounding.
>
> ---
>
> We sincerely thank the Reviewer Xh1p for the thoughtful comments and constructive suggestions :). Your insights have been invaluable in improving the clarity and completeness of our work. In response, we have added analyses on data efficiency and low-resource robustness (**Appendix B.6**), included statistical significance studies for the 33B model (**Appendix B.7**), and expanded our model merging investigation with parameter-space and representational similarity analyses (**Appendix F**). All corresponding revisions have been incorporated into the latest manuscript and highlighted in **blue** for clarity.
>
> If you have any further questions or concerns, please feel free to contact us at any time. We are always available and look forward to further discussions with you. :)
>
> Best regards,
>
> All Authors

---

### Author Response · Authors · 2025-11-27
**Summary of Revisions and Additional Experiments**

Dear Reviewers and ACs,

We sincerely thank you for your time and constructive feedback. We have carefully considered all comments and updated the paper, with changes highlighted in **blue**.

During the rebuttal period, we conducted extensive additional experiments and provided detailed clarifications to address reviewers' questions. **These additions validate and strengthen our original conclusions**:

### **Additional Experiments:**

> * **Statistical Significance Analysis at Scale:** We performed rigorous significance testing across 5 independent runs for the 33B model. (Reviewer Xh1p W1, Q3, **Appendix B.7**)
> * **Data Efficiency and Domain-Specific Scarcity:** We evaluated performance under severely limited data conditions and when algorithmic training data is entirely absent. (Reviewer Xh1p W2, Q2, **Appendix B.6**)
> * **LoRA Fine-Tuning Comparison:** We compared model merging against LoRA-based adaptation strategies. (Reviewer 66mV W1, **Appendix B.5**)
> * **Post-Merge Unfreezing Ablation:** We conducted ablations unfreezing different visual modules to assess cross-modal distribution shifts. (Reviewer FB3k W2, Q2, **Appendix B.4**)
> * **Threshold Sensitivity Analysis:** We provided a complete threshold sweep for InfiBench-V's pass/fail mapping. (Reviewer FB3k W1, Q1, **Appendix D**)
> * **Standalone Code LLM Evaluation:** We evaluated multiple Code LLMs on multimodal benchmarks to demonstrate the necessity of visual grounding and validate that gains stem from true multimodal fusion. (Reviewer Qdg3 W1, Q4, **Appendix B.8**)
> * **Specialized Model Comparison:** We compared with ChartCoder, a specialized chart-to-code model. (Reviewer Qdg3 Q1, **Appendix B.9**)
> * **Robustness Under Visual Degradation:** We evaluated performance on noisy inputs. (Reviewer Qdg3 Q2, **Appendix B.11**)
> * **Merge Coefficient Sensitivity:** We evaluated performance across a wide range of λ values. (Reviewer Qdg3 W2, Q4, Q6, **Appendix B.10**)
> * **Catastrophic Forgetting Assessment:** We evaluated general VQA tasks to assess whether model merging degrades visual understanding. (Reviewer Qdg3 Q2, **Appendix B.12**)

### **Clarifications:**

> * **Mechanistic Analysis of Model Merge:** We conducted parameter-space orthogonality analysis, layer-wise representational similarity analysis via CKA, and comparison with backbone replacement strategy to understand why model merging enhances multimodal coding capabilities. (Reviewer Xh1p W3, Reviewer 66mV Q1, Reviewer Qdg3 W2, Q4, Q7, **Appendix F**)
> * **Dataset Quality and Diversity:** We provided comprehensive statistics on chart categories and image quality distribution in InfiBench-V. (Reviewer Qdg3 W3, Q3, **Appendix E.3, Appendix E.4**)
> * **Training Stability Analysis:** We added smoothed training loss curves to demonstrate optimization stability after model merging. (Reviewer Qdg3 W2, Q6, **Appendix I.1**)
> * **Theoretical Foundations:** We provided theoretical justification for task-vector composition, discussing parameter-space orthogonality, linear interference theory, and cross-domain ability fusion. (Reviewer Qdg3 W2, Q4, Q6, Q7, **Appendix I.2**)
> * **HTML Generation Prompt Template:** We added the complete GPT-4o prompt used for HTML generation in MCD. (Reviewer Qdg3 Q3, **Appendix G.2**)

---

We thank all reviewers for their constructive feedback, which has substantially strengthened our work. We believe these extensive revisions thoroughly address all concerns raised and demonstrate the solid contributions of VisCodex to multimodal code generation.

Best regards,

All Authors

---

### Author Response · Authors · 2025-12-03
**Summary of the Rebuttal Period for Area Chairs**

Dear Area Chairs,

We sincerely thank all reviewers and ACs for their thorough review and timely engagement during the rebuttal process. In light of this year's unusual rebuttal process, we provide this concise summary to assist your assessment. All points are grounded directly in the rebuttal thread.

---

### **Key Strengths Recognized by Reviewers**

We are encouraged that reviewers consistently recognize the key strengths of our work:

* **Novel Problem and Solid Contributions** (Reviewers Xh1p, 66mV, Qdg3, and FB3k): The paper addresses a **meaningful and underexplored capability**—generating functional code from visual inputs—with **high practical value**. The work provides **a comprehensive solution with clear and methodologically sound presentation**.

* **Novel and Efficient Model Merging Approach** (Reviewers 66mV, Qdg3, and FB3k): The task-vector-based merging strategy integrates coding capabilities into vision-language models while **preserving both visual comprehension and advanced coding skills**.

* **Large-Scale, High-Quality Resources** (Reviewers Xh1p, FB3k, and Qdg3): MCD provides **598k meticulously curated samples** spanning diverse multimodal coding tasks. InfiBench-V offers a **benchmark with visually-rich programming questions** where images are indispensable.

* **Strong Empirical Performance** (Reviewers Xh1p, 66mV, Qdg3, and FB3k): Comprehensive experiments show **consistent improvements** over open-source baselines with **competitive performance against proprietary systems**. Extensive ablations **validate the effectiveness** of model merging for multimodal code generation.

---

### **Summary of Reviewer Responses**

To assist the Area Chair in decision-making, we summarize the positive consensus achieved during the rebuttal period. **Prior to the OpenReview leak incident, our submission effectively transitioned from mixed scores (6, 6, 6, 4) to acceptance-level consensus (6, 6, 6, 6), with Reviewer Qdg3 raising their score from 4 to 6.**

* **Reviewer Qdg3** (Score: 4 → **6**): The reviewer expressed that **"most of my concerns have been resolved"** and stated being **"mostly positive about this paper,"** raising their score to 6 (commented on 25 Nov 2025, 06:21 UTC, before the leak incident).

* **Reviewer 66mV** (Score: **6**): The reviewer explicitly stated that their **"primary issues are resolved"** and confirmed they **"maintain their positive rating of this paper"** at 6 (commented on 28 Nov 2025, 01:48 UTC, after the leak incident).

* **Reviewers Xh1p and FB3k** (Initial Scores: **6**, **6**): While these reviewers did not participate in the rebuttal discussion before the system freeze, both provided initially positive scores supporting acceptance. We have conducted substantial additional experiments and provided detailed clarifications to directly address their specific concerns:

  * **Reviewer Xh1p** raised concerns about: (W1) statistical significance of marginal gains at the 33B scale, (W2) limited exploration of data-scarce scenarios, and (W3) lack of deeper mechanistic analysis. We addressed these through rigorous significance testing across 5 independent runs showing statistically significant improvements (p < 0.05 across all benchmarks, **Appendix B.7**), comprehensive data efficiency experiments demonstrating that model merging maintains robust performance even with only 1% training data or complete absence of algorithmic data (**Appendix B.6**), and extensive mechanistic analyses including parameter-space orthogonality, representational similarity (CKA), and comparison with backbone replacement, demonstrating that model merging enables successful integration of code reasoning while maintaining visual grounding (**Appendix F**).

  * **Reviewer FB3k** requested: (W1/Q1) threshold sensitivity analysis for InfiBench-V's pass/fail mapping, and (W2/Q2) post-merge unfreezing ablations to assess cross-modal distribution shifts. We provided a complete threshold sweep (0-100) identifying the optimal threshold at 85 with 86.07% agreement with human annotations (**Appendix D**) and ablation studies unfreezing different visual modules, showing comparable performance across all configurations and confirming no detrimental cross-modal distribution shift (**Appendix B.4**).

  We believe these additional experiments and analyses effectively resolve their concerns. Given the positive strengths Reviewers Xh1p and FB3k recognized in their initial reviews and the comprehensive resolution of the raised concerns, we believe there is a **strong chance that their scores would increase.**

---

We provide a detailed summary of all additional experiments and clarifications conducted during the rebuttal period in the comment **"Summary of Revisions and Additional Experiments"** below.

We sincerely appreciate your time and consideration in reviewing our work during this unusual rebuttal process.

Best regards,

All Authors

---

### Meta-Review · Area_Chair_q85r · 2025-12-31

**Summary:**

This paper proposes VisCodex, a unified framework for multimodal code generation that integrates a vision-language model with a code-specialized LLM via a task-vector–based model merging strategy. In addition to the methodological contribution, the authors introduce a large-scale Multimodal Coding Dataset and a challenging benchmark targeting visually grounded, real-world programming tasks. The work is evaluated across a broad set of multimodal code generation benchmarks and demonstrates strong performance among open-source MLLMs, approaching proprietary systems such as GPT-4o.

Overall, reviewers agree that the paper addresses an important and timely problem, provides a technically sound and well-presented solution, and offers substantial empirical evidence to support its claims. The rebuttal and revisions significantly strengthened the paper by addressing nearly all raised concerns with additional experiments and analyses.

**Reviewer Concerns:**

After rebuttal, reviewers who actively participated in the discussion explicitly stated that their primary concerns were resolved and maintained or raised their positive evaluations. No substantial unresolved issues remain.

**Reviewer Scores:**

Based on the rebuttal discussion:

- Reviewer Qdg3 explicitly increased their score from 4 to 6 after concerns were resolved.

- Reviewer 66mV confirmed that their primary issues were resolved and maintained a positive score.

- Reviewers Xh1p and FB3k initially provided borderline-accept scores; given the extensive additional experiments directly targeting their critiques, it is reasonable to expect their evaluations would remain positive or improve if full discussion had been possible.

Overall, the post-rebuttal consensus supports acceptance.

---

### Decision · Program_Chairs · 2026-01-26

Accept (Poster)